



# 4D dispersion of total gaseous mercury derived from a mining source: identification of criteria to assess risks related with high concentrations of atmospheric mercury.

José M. Esbrí[1], Pablo L. Higueras[1], Alba Martínez-Coronado[2], Rocío Naharro[1].

[1]Instituto de Geología Aplicada, Castilla-La Mancha University, Almadén (Ciudad Real), 13400, Spain
[2]Asociación española contra el Cáncer, Ciudad Real, 13002, Spain

*Correspondence to*: José María Esbrí (josemaria.esbri@uclm.es); Pablo L. Higueras (pablo.higueras@uclm.es)

**Abstract.** Mercury is a global pollutant that can be transported long distances after its emission by primary sources. Most of the problems associated with Hg as a toxic element dispersed worldwide arise due to its incorporation into the trophic chain and its

conversion into organic forms. However, in the vicinity of anthropogenic sources, the most common problem is the presence of Hg in inorganic forms and in the gaseous state in the atmosphere. Risk assessments related to the presence of gaseous Hg in the atmosphere at these contaminated sites are often based on episodic and incomplete data, which do not properly characterize the Hg cycle in the area of interest or consider spatial or temporal terms. The aim of the work described was to identify criteria to obtain the minimum amount of data with the maximum meaning and representativeness in order to delimitate risk areas, both in a spatial

and temporal respect. Data were acquired from September 2014 to August 2015 and included vertical and horizontal Hg measurements. A statistical analysis was carried out and this included the construction of a model of vertical Hg movements that could be used to predict the location and timing of Hg inhalation risk. A monitoring strategy was designed in order to identify the relevant criteria and this involved the measurement of gaseous Hg in a vertical section at low altitude (i.e., where humans are present) and in horizontal transects to characterize appropriately the transport cycle of gaseous Hg in the lower layers of the

atmosphere. The measurements were carried out over time in order to obtain information on daily and seasonal variability. The study site selected was Almadenejos (Ciudad Real, Spain), a village polluted with mercury related to decommissioned mining and metallurgical facilities belonging to the Almadén mercury mining district.

The vertical profiles revealed that higher Total Gaseous Mercury concentrations are present at lower altitude during nocturnal hours and at higher altitude at dawn and dusk. Horizontal profiles showed that the background values were close to 6 ng m$^{-3}$ except in the

spring months, when they rose to 13 ng m$^{-3}$ and increased the area affected by mercury emissions to more than 4 km around the mining and metallurgical sites. On a daily basis the most important process involved in gaseous mercury movements is the mixing layer, which begins in the early morning and finishes at nightfall. Vertical transferences are predominant when this process is active, i.e., in all seasons except winter, while major sources act as constant suppliers of gaseous Hg to the mixing cell, thus producing Hg deposition at dusk. Conversely, horizontal transferences prevail during the hours of darkness and the main factors are major and

minor sources, solar radiation, wind speed and topography.

In terms of risk assessment, and based on the model constructed to infer atmospheric Hg concentrations based on micrometeorological parameters, the nights carry greater risk than the days in all seasons (54% in spring and winter, 72% in summer) except in autumn, when 99% of the hours of risk occurred during the day. The main factors involved in the creation of high-risk periods are those related to dilution (or its absence): namely wind speed and solar radiation at null levels. The extent of the area

affected by an emission source is independent of its importance in terms of absolute emissions. The affected zone did not extend beyond 100 metres from the location of the source during the daytime period and 200 metres in the night-time. Under the worst micrometeorological conditions, it was predicted that the affected area would cover almost the entire town of Almadenejos, although these risk conditions only represent 11.34% of the hours in an annual period.



The results of this study highlight the possible importance of the relief in the distribution of gaseous mercury in the proximity of
discrete sources. Further studies, including a detailed topographic model of the area, are required in order to make precise estimations
of the influence of this parameter, which appears in this study to be less important than the other factors but is still appreciable.

## 1 Introduction

Mercury (Hg) is considered to be a global pollutant due to its ability to be transferred between different environmental compartments
and over long distances, which results in the contamination of pristine areas far from the sources. The Hg cycle in the environment
begins with geogenic or anthropogenic emissions, which mainly consist of gaseous elemental mercury (GEM) along with minor
proportions of particle-bound mercury (PBM) and reactive gaseous mercury (RGM). Together these species constitute 'total gaseous
mercury' (TGM). The residence time of each of these mercury species is different and is much longer for GEM, which means that
this species can be deposited in remote areas such as the Arctic Sea, while PBM and RGM are often deposited on local or regional
areas that are relatively close to the source (Radke et al., 2007, and references therein). Once Hg is being deposited, a cycle of re-
emission/deposition, along with changes in Hg speciation, explains the flows of this element in the environment.
Numerous Hg transfer pathways are involved in this cycle, and these include soil-atmosphere, soil-plant, plant-atmosphere, and
water-atmosphere, amongst others. These fluxes have been quantified by different approaches, most of which employ dynamic flow
chambers, micrometeorological methods and bulk methods (Carpi and Lindberg, 1997; O'Driscoll et al., 2003; Stamenkovic et al.,
2008; Eckley et al., 2011; Zhu et al., 2015; Fu et al., 2016; Zhu et al., 2016, amongst others). Some doubts remain concerning the
comparison of these methods, but the results show that processes of Hg deposition and emission are included in a complex cycle in
which very different factors influence the flows from one environmental compartment to another and a large number of factors are
involved between these, mainly seasonality, vegetation coverage, temperature, solar radiation, relative humidity, diurnal
atmospheric turbulence and the presence of Hg oxidants (Zhu et al., 2016). A maximum emission during diurnal hours was described
for soils (Zhu et al., 2015), mine materials (Eckley et al., 2011), waters (O`Driscoll et al., 2003) and snow (Maxwell et al., 2013),
while forb leaf (Stamenkovic et al., 2008) and growing broad leaf (Fu et al., 2016) reach their minimum emission rates during
diurnal hours. These daily cycles of Hg emissions from soils, waters or plants contribute to the increase of the atmospheric mercury
pool, especially in the lower layers of the troposphere. Most of the available information on this topic is on a kilometric scale, at
altitudes in the range 500–11,000 metres from background and contaminated locations; however, information about TGM dispersion
on a metric scale is scarce. Some information about these distances comes from episodic monitoring by means of LIDAR techniques,
such as those measured in China, where maximum levels at lower altitudes were detected during night-time hours (Guan et al.,
2010). Saiz-Lopez et al. (2008) modeled the vertical profile of GEM over Antarctica and found that maximum levels were located
at lower altitudes during daytime hours. Tackett et al. (2007) described a vertical GEM profile in the Arctic troposphere and found
maximum levels of GEM at heights of 20–80 metres above ground under different conditions. Steffen et al. (2002) studied vertical
profiles on snowpack before and during depletion events and found that GEM levels increased sharply at the surface during the
depletion event on a two-metre profile. Ferrara et al. (1998) identified higher TGM concentrations a few centimetres above the
ground and background values at heights 10–20 metres above ground at the Eastern border of Almadén village.
Risk assessments of areas with anthropic contamination of gaseous Hg are often carried out with scarce data, often corresponding
to short periods of time, and these do not provide a representative view of the day-night contrast or the seasonality, not even at the
level of hot and cold or dry and wet seasons (depending on the location of the case study). We have conducted studies based on
sampling in the worst theoretical conditions with the aim of identifying the worst-case scenario (Martinez-Coronado et al., 2011;
Vaselli et al., 2013; Esbrí et al., 2015; 2018a; Tejero et al., 2015), the evaluation of background conditions (Higueras et al., 2014),
and comparison of the worst and best scenarios (Higueras et al., 2013). However, we realized that the mercury cycles in all of the
studied sites were not exactly the same, and that the most important factors that control the emission, transport and deposition
processes also differed from one area to another. This idea can be exemplified by the most recent reference found for risk assessment



related to gaseous mercury (Deng et al., 2016). The authors found correlations between methylmercury (MeHg) in blood and TGM in the air after only seven gaseous Hg measurements without covering a full annual period. There are many confounding variables that may produce this correlation due to the low representativeness of the gaseous Hg sampling. It therefore seems necessary to carry out a sufficiently representative sampling effort in order to understand the peculiarities of the Hg cycle in the study area to achieve a realistic risk assessment. In this sense, it is necessary to gain a knowledge of the Hg flows from the moment that it is

emitted and to understand the main mechanisms of dilution and/or concentration. It is also necessary to obtain information on the evolution in the vertical direction with respect to the source and also horizontally, as this information that will provide the dispersion of the contaminant in the area. It is also highly recommended to obtain information on the temporal evolution of these vertical and horizontal processes, both daily and seasonally.

The main objective of the work described here was to obtain information on vertical and horizontal profiles of TGM in an

environment contaminated by decommissioned mercury mining facilities, with the ultimate objective of locating the risk areas around the main sources of emission and the moment at which the risk in these areas was significant. Data acquisition was carried out over a whole year in order to identify relationships between TGM data and secular variations in local micrometeorological and topographical data. In this way, the title of the manuscript refers to a new type of 4d monitoring, in the sense that Wikipedia defines the term: "meaning the 4 common dimensions, is an important idea in physics referring to three-dimensional space (3D), which

adds the dimension of time to the other three dimensions of length, width, and depth".

## 2 Methodology

In this work we have tried to obtain the minimum information necessary about the emission, transport and deposition of atmospheric mercury to ensure the representativeness of such data with a minimum cost in terms of effort and money. Before designing the sampling locations, an exhaustive identification of the Almadenejos emission sources, represented in red in Fig. 1, was carried out.

In the town centre of Almadenejos there are four emission sources of medium importance, while in the vicinity there is one of very high importance (MMP), one of high importance (Nueva Concepción mine), and two of low importance (a contaminated road running North of the town and the course of the Valdeazogues river, since it passes through the El Entredicho mine).

In an effort to achieve the main objective of this work, it was decided to obtain data in the three directions of space, including a short vertical transect and long horizontal transects that include emission sources of high, medium and low importance. These data

were also obtained serially over time to cover both the daytime cycles of light and darkness, as well as the seasonal cycles of hot and cold periods.

### 2.1 Vertical profile measurements

Total Gaseous Mercury (TGM) was measured at a site located in the proximity of Almadenejos village (Fig. 1) and corresponding to a closed precinct that encompasses the Almadenejos wastewater treatment plant (AWTP) (WGS84 30S 351707 E/4289235 N).

Almadenejos was a secondary mining and metallurgical centre in the so-called Almadén mercury-mining district (Higueras et al., 2006). This area includes three large mines, which are now closed, and a metallurgical precinct located immediately to the North of the urban area and representing the only significant active local source of gaseous mercury (Martínez-Coronado et al., 2011). This is an excellent area for the study of Hg transference between environmental compartments (Naharro et al., 2018; Campos et al., 2018; Esbrí et al., 2018b) due to the scarce remediation works that have been carried out in recent centuries, which has left a legacy

of anomalous Hg presence in the soils, roads and rivers. This situation gives rise to a very interesting mercury cycle to carry out environmental studies.

The equipment used to make the measurements was a Tekran 2537B with a synchronized multi-port sampler (model 1115) that allowed alternate measurement of up to six separate input streams. In this work only three of these six sampling possibilities were used, and these corresponded to sampling points located at 0.5, 2 and 3 metres above ground, with measurements made from





September 2014 to August 2015. An external pump working at 15 m$^3$ h$^{-1}$ was employed, with the sampling lines purged with Hg-free air (GEM < 2 ng m$^{-3}$) when not in use. Various gradients were used in order to study differences in TGM contents at different altitudes: Total Gradient (TGM$_{3metre}$–TGM$_{0.5metre}$), Lower Gradient (TGM$_{3metre}$–TGM$_{2metre}$) and Upper Gradient (TGM$_{2metre}$–TGM$_{0.5metre}$).

The device was calibrated every seven days by means of an internal permeation source. An intercomparison exercise between Lumex

RA-915M and Tekran 2537B systems was carried out in 2011 in conjunction with the Spanish *Instituto de Salud Carlos III*, and a compatibility index (see ref. ISO/IEC, 1997) of less than 1 was found during all experiments (Fernández-Patier and Ramos-Diaz, 2011).

Micrometeorological data were acquired using a Davis Vantage Pro meteorological station, which is a fully automated device that allows data to be collected every 15 minutes, including temperature, relative humidity, wind speed and direction, atmospheric

pressure, rain, solar radiation and ultraviolet radiation. The location of this device (WGS84 30S 351714 E/4289255 N) is shown in Fig. 1.

**Figure 1: Location of the three horizontal profiles in the study area and main mercury mining sites. AWTP: Almadenejos Wastewater Treatment Plant. MMP: Mining and metallurgical plant. Main known gaseous mercury sources are shown in red (including rivers with**

**Hg-contaminated sediments); uncontaminated streams are shown in blue.**



## 2.2 Horizontal profile measurements

The sampling strategy was designed to cover not only anomalous GEM data from local emission sources, but also background GEM data around major and minor sources at locations far from Almadenejos. As shown in Fig. 1, three transects were chosen to achieve these objectives:

- Profile 1 has a length of 12,350 metres, including two crosscuts with the Valdeazogues River, which transports sediments moderately contaminated with Hg (García-Ordiales et al., 2016), especially in the vicinity of El Entredicho open pit mine (Fig. 1). This profile passes close to Almadenejos village, 40 metres from the main GEM source in the studied area, which is a decommissioned mining and metallurgical plant (MMP) (Martínez-Coronado et al., 2011). This profile was selected with the aim of identifying relationships between background GEM data, at large distances from Almadenejos, and local anomalous GEM data from the main source area.

- Profile 2 has a length of 3650 metres and it crosscuts the main GEM sources of the village: La Nueva Concepción mine, MMP and two minor GEM sources in tracks paved with contaminated materials from the MMP. This profile was selected with the aim of identifying possible relationships between anomalous GEM data from local sources and background data from the hills located to the South and North of Almadenejos.

- Profile 3 covers 8,450 metres from Almadenejos village limits to the village of Gargantiel. This profile mainly represents the background values, including a crosscut over two minor GEM sources of polluted sediments: the Valdeazogues River and its tributary, the Gargantiel River.

Data acquisition of GEM was performed using a Lumex RA-915M device, which is a portable Atomic Absorption Spectrometer that is able to collect one GEM data point every second (Higueras et al., 2014). The device was installed on an automobile, with the sampling line located on the front side of the vehicle. The speed was kept constant during the sampling time, i.e., in the range 40–50 km h$^{-1}$. Baseline checks were carried out at the beginning and the end of each profile, and a baseline correction was performed when differences were up to 1 ng m$^{-3}$, assuming that the lamp derive was linear during measurements. The profiles had differences in the number of data points and precise locations along the profile. In order to enable direct comparison of the collected data and to minimize erraticism, an average for each 100 metres was calculated and each average was assigned to the centre of the corresponding distance range.

## 2.3 Statistical treatment

Data analysis was carried out with different software packages: Microsoft Excel, Minitab 15 and Golden Surfer 9. A multiple linear regression analysis (MLRA) on the normalized dataset of vertical profiles was performed using Minitab 15. A best subset regression analysis was performed using Mallows' CP to identify the best predictors prior to performing a multiple linear regression analysis on each dataset. A fitted line graph was constructed using the equation obtained in the MLRA to obtain an R$^2$ value based on a new equation between measured gradient (or TGM) and predicted gradient. Lepeltier graphs were used to find the distribution pattern that best fitted the various sets of GEM data in horizontal profiles. A lognormal distribution curve (Lepeltier, 1969) is defined by two parameters: one is dependent on the mean value and the other is dependent on the character of the distribution of values. These parameters were determined graphically by means of cumulative frequency curves in log-probability plots using Minitab 15. Finally, the delimitation maps of risk areas due to the presence of gaseous Hg were produced using Surfer 9.





## 3 Results and discussion

### 3.1 Vertical profiles

The daily evolution pattern of TGM (Fig. 2A) is similar to that described by Esbrí et al. (2016) for Almadén town, which is located
11 km to the west of Almadenejos: the pattern shows low TGM levels during diurnal hours and higher levels during the rest of the
day, a finding that has been interpreted as being due to a decrease in the wind speed during the night (Fig. 2B). In terms of TGM
levels (see Table 1 and Fig. 2C for more details), the site studied in Almadenejos has TGM concentrations that are at least three
times higher than those measured in the Almadén site described by Esbrí et al. (2016) and a more marked difference between
maximum and minimum daily TGM concentrations was observed. This behaviour is probably due to the fact that in Almadén the
main dump was reclaimed during the years 2008–2010, which led to a very significant decrease in local TGM (Higueras et al.,
2013), whereas reclamation of the metallurgical precinct has never been performed in Almadenejos. As a result, a huge amount of
metallurgical waste remains in the MMP, in addition to the presence of some minor sources produced by the network of roads and
tracks or uncontrolled accumulation of contaminated waste (Fig. 1). Seasonally, the main pattern is similar to that described by
Esbrí et al. (2016) in the nearby population of Almadén, with lower TGM levels in winter and higher levels in summer. However,
transitional seasons show a different trend, especially in springtime, when TGM levels are at an intermediate level between winter
and autumn.

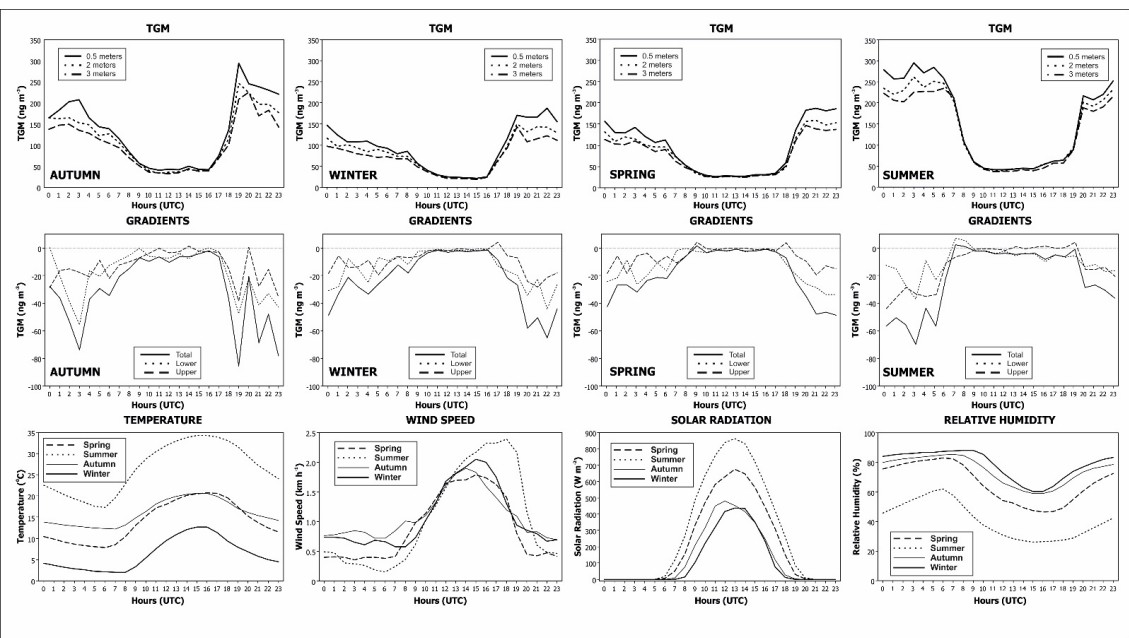

**Figure 2:** Daily and seasonal evolution of TGM contents at 3, 2 and 0.5 metres above ground in the AWTP site (Upper row); daily and
seasonal evolution of gradients at between 3–0.5 metres, 3–2 metres and 2–0.5 metres above ground in the AWTP site (Middle row); and
**daily and seasonal evolution of micrometeorological parameters (temperature, solar radiation, relative humidity and wind speed) in the
AWTP site (Lower row).**

Vertical gradients in the AWTP (Table 2 and Fig. 2) show TGM concentrations close to zero during diurnal and windy hours, with
higher concentrations at lower heights above ground in all three sections considered (3–0.5; 3–2; 2–0.5). Variability according to
season does not appear to have any evident pattern, except for summer versus autumn/winter/spring: Summer data reached positive
levels only in the first few hours before and after the atmospheric mixing process was active, i.e., at dawn and dusk, especially for
gradients 3–0.5 and 2–0.5. These positive differences between heights in terms of TGM indicate that mercury remains accumulated
at lower heights during the night, it rises while the mixing layer is being created, and it falls when this mixing layer disappears.
These data indicate that a diurnal cycle of emission and deposition is active in the studied area, and that deposition is intense –

especially at dusk – in the transitional hours between higher and lower winds. Llanos et al. (2011) estimated the annual emissions

of the MMP to be 16.4 kg y$^{-1}$, but the dispersion/dilution process of these emissions in the surrounding environment was unknown
and it was impossible to perform a similar measurement with a crane over this MMP facility.

The daily evolution of the maximum, medium and minimum values for each of the three heights monitored is represented in Fig. 3, with the aim of visualizing the vertical movements of Hg and the heights at which they occur, both ascending and descending. It can be observed that in the summer the flows in the highest monitored sector occur only in the early hours of the morning, thus

precluding the movement of Hg from the lowest to the highest monitored height. Instead, the exchange in the lower part is continuous throughout the day, i.e., from dawn to dusk. This phenomenon must be due to the confluence of three micrometeorological factors: high temperatures and solar radiation coincidental with low relative humidity values, which combine to increase the intensity of the formation of the mixing layer during the day. In spring a similar exchange of maximum and minimum values also occurs early in the day, but in this case the exchange is maintained throughout the day only in the highest monitored sector. The main difference

between these two seasons is the soil temperature, which is much lower in spring (16 ℃) than in summer (27 ℃). In the spring, the lower soil temperature would promote a thinner mixing layer, which would be unable to promote Hg transfers in the region close to the ground, while in summer the high ambient and soil temperatures would increase the thickness of the mixing layer, thus producing Hg transfers in lower areas. Autumn, the other transition season, shows flows in the upper sector during the day and in the lower part only at dawn, when wind had ceased. Finally, in winter, when micrometeorological factors attenuate the creation of the mixing

layer, flows were not detected during the day and only at dusk was a single exchange in the upper part of the monitored sectors measurable.

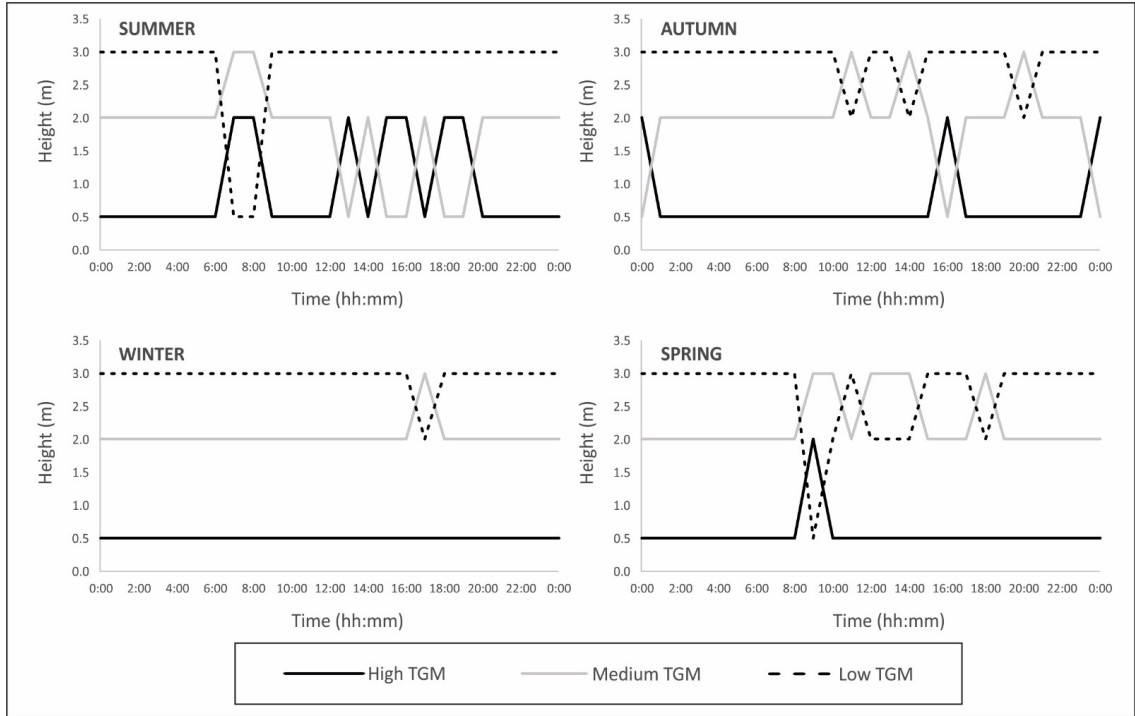

**Figure 3: Schematic representation of the daily evolution of the maximum, medium and minimum levels of gaseous Hg at the three measuring points in the vertical profile.**

The evolution of the TGM gradient during a typical summer day (with symmetrical temperature and solar radiation profiles during daytime hours) exemplifies this process perfectly (Fig. 4a), with positive gradients observed as solar radiation declined and an increase in the early morning. Negative gradients were observed during night-time hours, probably as a consequence of the stratification of lower atmospheric layers when the wind speed was zero or close to zero. A similar trend was observed on a winter



day (Fig. 4b) or on a day with thermal inversion in the morning (Fig. 4c), but this tendency was not observed during rainy periods,

such as a rainy day in November (Fig. 4d), on windy days (Fig. 4e) or on misty days (Fig. 4f).

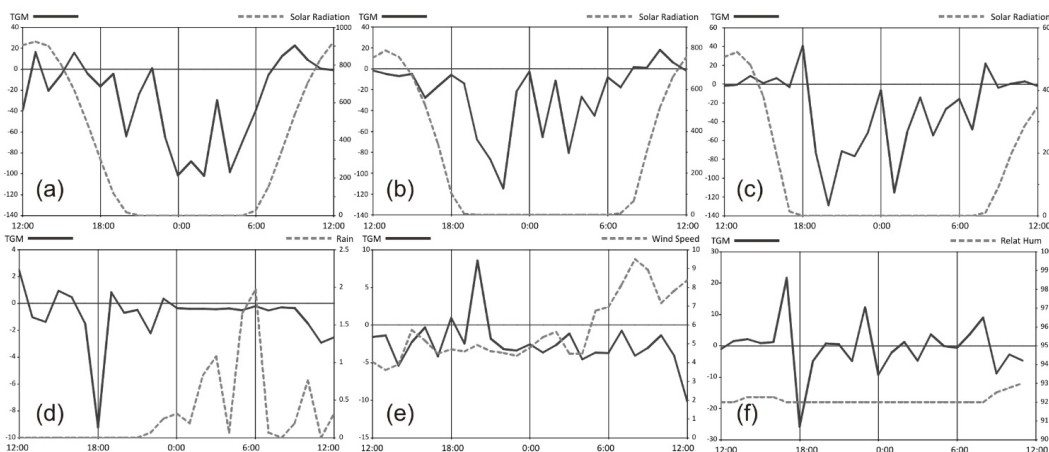

**Figure 4: Evolution of TGM total gradients during typical contrasting meteorological conditions: (a) typical summer day; (b) typical winter day; (c) thermal inversion day; (d) rainy day; (e) windy day; and (f) foggy day.**

On considering the weight of each factor by applying a multiple linear regression analysis (Table 3), it was observed that solar

radiation is the key factor in spring, autumn and winter, but that wind speed can better explain gradient data in summer. Surprisingly, temperature appears to be a secondary factor in all seasons, and only in the summer period and in the lower gradient does this appear to be an important factor. The identification of wind speed as a primary factor in the lower gradient in autumn is consistent with the Hg exchange found at dusk in this period (Fig. 3). Wind speed is also important in the upper gradient in winter, i.e., the sector where the exchange of Hg is appreciable in Fig. 3.

**3.2 Horizontal profiles**

Profile 1 (Fig. 5a) represents the longest transect (12,350 m) and it includes two Valdeazogues river crosscuts: one near the El Entredicho open pit mine (located 2,500 metres upstream) and the other at the beginning of the transect (10,800 metres downstream from El Entredicho). Higher GEM contents were found in summer and spring, especially in the Valdeazogues river crosscut closest to El Entredicho. In terms of data variability, transition seasons (spring and autumn) show the highest differences between maximum

and minimum GEM concentrations, while winter presents the lowest term. This data variability reaches its maximum amplitude in anomalous values (up to 60 ng m$^{-3}$ during summer and spring near the MMP). Furthermore, data variability shows clear differences in background values between seasons, i.e., highest for transition seasons and lowest for summer and winter. Background levels are close to 6 ng m$^{-3}$ during all seasons except for spring, which gave a value of 13 ng m$^{-3}$. Springtime is characterized in Almadenejos by a marked increase in temperature and solar radiation (Fig. 2), as compared with previous winter months, and an intense soil

moisture release occurs that could enhance soil mercury emissions through the volatilization of more labile soil mercury species (Llanos et al., 2011). This process significantly increases the extent of higher background GEM levels, thus increasing the area affected directly by mercury emissions to more than 4 km taking into consideration the distance in profile 1 (Fig. 5a) from the Valdeazogues River to the MMP.



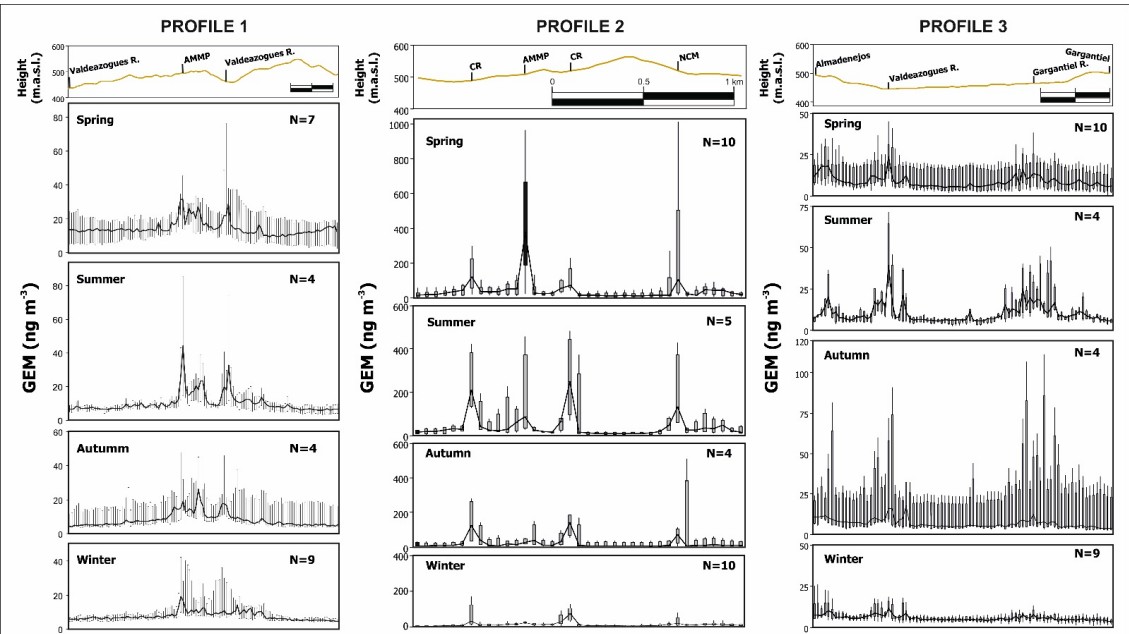

Figure 5: Boxplot of GEM in the horizontal profiles along the Almadenejos area. Each box represents the average of 100 metres. The location of these profiles is shown in detail in Fig. 1.

Profile 2 passes through minor and major mercury sources in the studied area, namely an abandoned metallurgical precinct (MMP), a closed underground mine (NCM) and two points with contaminated roads (CR in Fig. 5b). The maximum GEM concentrations were found during spring months, especially in the proximity of the MMP and NCM, and the lowest concentrations were measured during winter months (Fig. 5b). The extent of the anomalous high values reaches maximum distances in summer, with higher data variability along the transect, and minimum distances in winter, when anomalous values appear within the background values.

Profile 3 represents the local background profile and it crosses only one minor GEM source, the Valdeazogues River, 5,000 metres downstream from the El Entredicho mine (Fig. 5c). Maximum levels and variability in this transect were observed during autumn months, while winter represents the minimum for these two aspects. Background values were slightly lower (4–5 ng m$^{-3}$) for each season considered when compared to those for profile 1, although this profile represents a low-grade contaminated area in the mining district, with mercury present in sediments of the Valdeazogues and Rivera de Gargantiel Rivers (García-Ordiales et al., 2016), in soils (Rodríguez et al., 2003) and incorporated into the road as polluted waste from the El Entredicho closure works.

The Valdeazogues River represents a strip of contaminated materials that comprises the alluvial plain, and it is probably the most important minor source of GEM in the region. This river crosses the district for 30 km from the easternmost Hg mine (El Entredicho) to the confluence with the Guadalmez River outside the mining district. The GEM levels for all seasons in a section of the Valdeazogues River affected mainly by El Entredicho wastes are represented in Fig. 6 and it can be seen that the GEM average and range decrease with increasing distance from the mine.





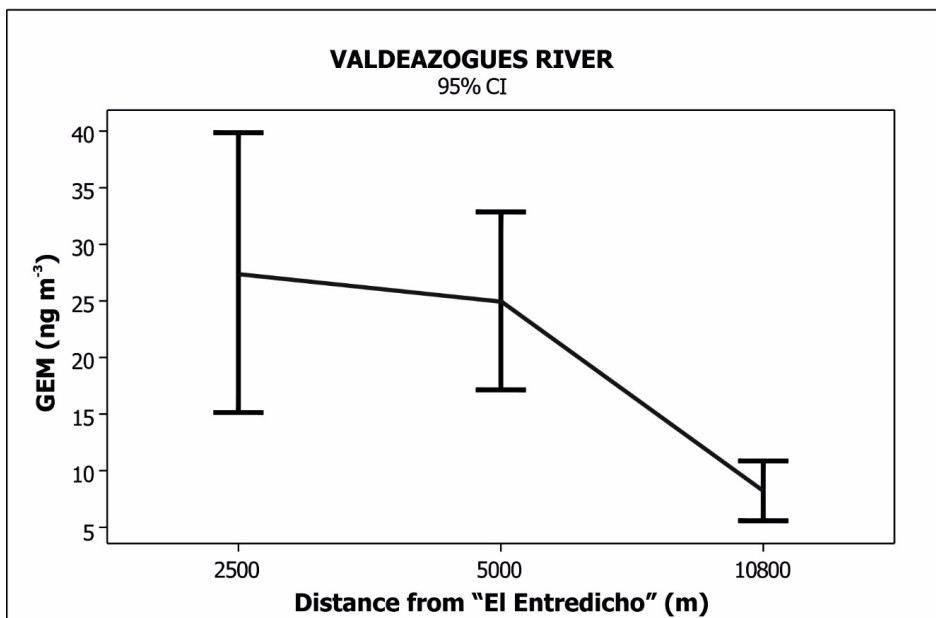

**Figure 6: Variability in GEM levels on crosscuts of the Valdeazogues River according to distance from El Entredicho mine.**

The profiles dataset was represented as lognormal distribution curves (Fig. 7) in an effort to determine the evolution of the main GEM sources in the studied area. The idea behind these graphs was to separate dataset populations, with the Y axis representing cumulative Gaussian distributions. The changes in the slope of each probabilistic curve (breaks) mark the boundaries between subpopulations. Profile 1 shows clear differences between 'classic' seasons and 'transition' seasons in terms of limits between normal and anomalous populations. In this sense, in summer and winter there is a break between normal and transitional populations

at similar levels (6.76 ng m$^{-3}$ in winter and 7.73 ng m$^{-3}$ in summer), while in spring this break occurs at higher levels (14.81 ng m$^{-3}$) and autumn it occurs at 10.36 ng m$^{-3}$. In profile 1 the anomalous population corresponds to the emissions of Valdeazogues riverbank sediments and these are detectable at Hg levels of 10 ng m$^{-3}$ in drier seasons (autumn and summer) and up to 30.14 ng m$^{-3}$ in wet seasons (winter and spring); it should be remembered that this profile does not have any significant emission sources, except for polluted sediments, and the background values are below 10–14 ng m$^{-3}$ in all seasons. A scenario in which emission sources are

absent is best represented by profile 3, since it only has two points of contact with contaminated sediments (3.5 kilometres away from the nearby El Entredicho mine) and therefore its total Hg contents are much lower than in profile 1. Normal values appear in profile 3 at around 5 ng m$^{-3}$ in summer and winter, and at around 10 ng m$^{-3}$ in spring and autumn. These values can be considered as the local geochemical background values for the study area. It should be noted that a second transition population appears in autumn and winter and this does not have an obvious explanation based on the appearance of a second source of emission in this

background value profile.

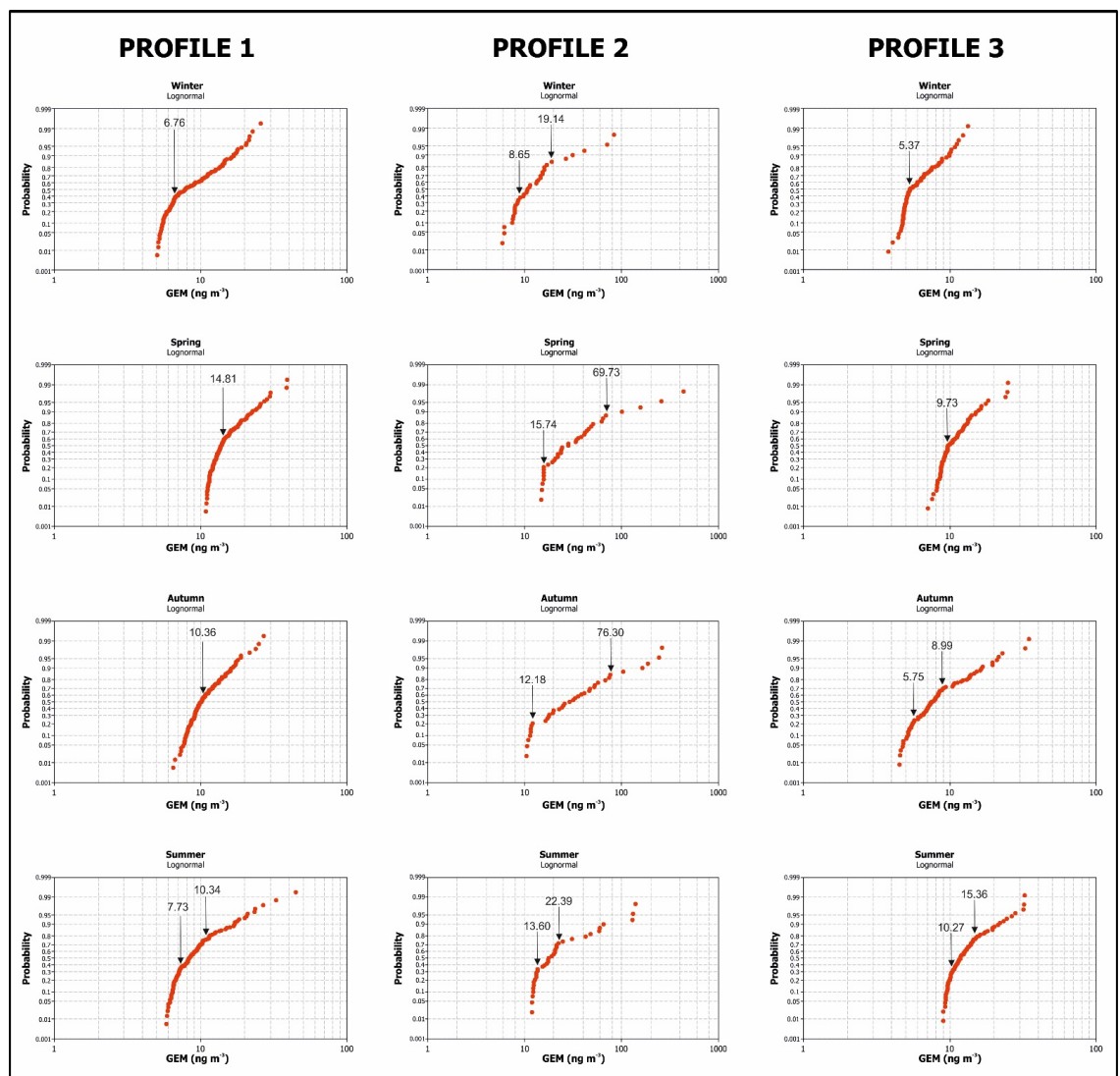

**Figure 7: Lepeltier graphs for each profile according to the season.**

Profile 2 has two emission sources of medium importance and one of high importance that produces an increase in the limit values of the normal population (around 15 ng m$^{-3}$ in all seasons except winter, which has a value of 8.6 ng m$^{-3}$). This scenario with multiple emission sources leads to the appearance of a second transition population in winter, autumn and summer. The existence of an emission source as important as the MMP produces an increase in the limit of anomalous values from around 70 ng m$^{-3}$ in winter and spring to 243 ng m$^{-3}$ in summer, i.e., well above the WHO (2000) limit for chronic exposure of 200 ng m$^{-3}$.

All datasets measured for these three profiles correspond to a period (11:00–14:00) of stability in terms of micrometeorological parameters, i.e., in the middle of the day, but we also offer data measured during the night in summer because Esbrí et al. (2016) and Tejero et al. (2015) reported higher GEM levels during nocturnal hours in the surroundings of mining-related GEM sources, with levels more than two times higher in Almadén, for instance. The present study provides information about the extent of this increase in nocturnal GEM levels. The evolution of Hg contents at three different time points, during the day when the wind ceases, and the probabilistic graph for each dataset are presented in Fig. 8. In profile 2 the maximum values are observed at the end of the monitoring, after the cessation of the wind, in all of the sources considered except for a contaminated road. This finding indicates

that the wind is a determining factor for the increase in the environmental concentrations of Hg despite the fact that the emissions are probably lower. The evolution of the increase in the GEM values from the beginning to the end is difficult to observe in the distance vs GEM graph, but the Lepeltier graph (Fig. 8B) offers a better view of the situation for all of the data populations. In profile 2 the shift towards higher GEM values occurs only in the transition population and not in normal or anomalous values. The reason for this increase in the medium values (transition population) but not in normal or anomalous populations could be explained

by the absence of dispersion processes, with background values and anomalous values remaining at similar levels because they are more dependent on emission factors (essentially temperature). So, what happens in profiles that do not have important emission sources? In profile 3 there are clearer increases towards the end of the monitoring without exception, and the increases in GEM contents affect all populations, i.e., normal, transitional and anomalous data. This trend provides evidence that the nocturnal increase in GEM values is homogeneous in the absence of significant sources of emission and only the topography seems to be an important

factor that drives this process, as can be seen in Fig. 8C: gaseous mercury emitted by minor sources tends to move downslope and becomes more dilute during this dispersion process. In the case of profile 3, this accumulation process happens in the bottom of the valleys. The Valdeazogues River and its sediments are a minor Hg source and the extent of higher nocturnal GEM levels reaches more than 1000 metres around the riverbed through a combination of both accumulation process, i.e., local emissions from sediments and topography.

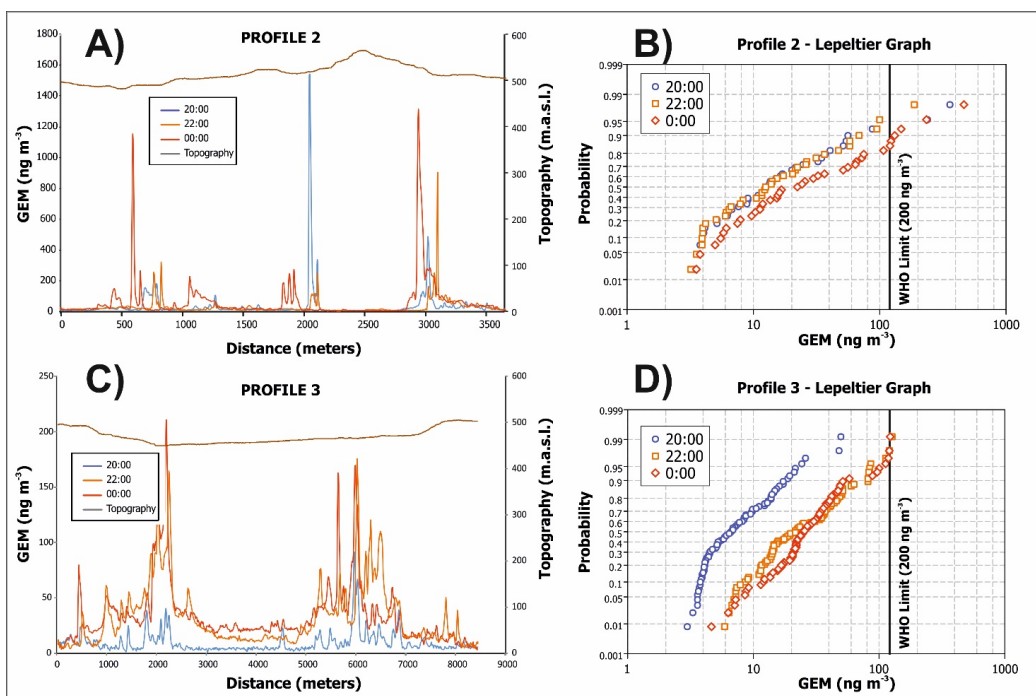


**Figure 8: Nocturnal GEM levels in Almadenejos and surroundings. Summer average represents the average of diurnal GEM measurements in summer surveys. Graphs B and D are the Lepeltier curves of datasets A and C, respectively.**

### 3.3 4D dispersion of TGM in the Almadenejos area

The preliminary conclusions from the monitoring work on the vertical and horizontal gradients suggest that dilution processes are

the key for explaining the movements of Hg in a mining-metallurgical environment with multiple emission sources scattered throughout the study area. The creation of the mixing layer in the early morning hours represents an increment in Hg dilution and this is driven by increases in solar radiation, temperature and winds, which simultaneously lead to enhanced mercury emissions due to the same factors (Carpi and Lindberg, 1997; Gustin et al., 2002; Lindberg et al., 1999). Major gaseous Hg sources act as a constant supplier to the mixing layer, thus promoting movements to the closest areas and the deposition of this mercury when the convective





forces cease and the mixing layer disappears. In terms of risk assessment, the existence of the mixing layer dilutes gaseous mercury and prevents the appearance of TGM levels up to 200 ng m$^{-3}$ in populated areas, thus restricting the zone affected to some tens of metres, even with a huge gaseous mercury source (MMP) very close to the houses in Almadenejos. Conversely, this process causes the dispersion of a large amount of mercury to the surroundings and this increases the risk of air-plant transfer, even to edible vegetables grown in local orchards. The most favourable seasons to activate this process are the driest and warmest (i.e., spring and

summer), while the micrometeorological conditions that can inhibit this process are rain, regional winds or persistent fog on winter days.

An opposite scenario occurs when local winds cease in twilight hours: emission rates decline with the absence of solar radiation and the decrease in temperature, while a wind speed close to zero produces an increase of TGM concentrations and strong negative gradients (Fig. 2) at human heights (from 0 to 3 metres above ground). Once again, summer is the season that has the most negative

gradients and this is due to the contrasting differences in micrometeorological parameters (Fig. 2). During these nocturnal hours, horizontal movements play the most important role in the transference of gaseous mercury in the area. Clear differences in processes between diurnal and nocturnal hours can be highlighted: dispersion/diffusion, dilution/concentration and the predominance of vertical/horizontal transferences. As a consequence, the extent of the areas affected by higher TGM levels increases, and it is not only the well-known major sources that play an important role, but also minor sources, which produce extended areas with more

than 200 ng m$^{-3}$ during summer nights without winds. Contaminated sediments of the Valdeazogues River act as the main secondary source in the Almadenejos area. This river receives polluted fine-grained mine materials, particularly from the El Entredicho mine (Fig. 1) but also from other nearby old mines such as Las Cuevas, Vieja Concepción and Nueva Concepción.

In terms of risk assessment, the monitoring strategy presented in this work was able to identify the main zones of the urban area that can reach TGM concentrations above the WHO limit for chronic exposure (200 ng m$^{-3}$) and also the period of time during the day

or throughout the year in which this limit is exceeded. The vertical gradients dataset obtained in the immediate vicinity of a medium emission source (AWTP) seems to indicate that, in terms of average values, only the night-time periods in summer and autumn produce concentrations above the WHO limit. To identify the hours during which inhalation of gaseous Hg may occur, one must take into account the whole dataset and not only these average values. Likewise, we identified the most favourable conditions for the WHO limit for chronic exposure to be exceeded (Table 4). This was achieved using the MLRA equations for the dataset at 2

metres (Table 3), i.e., the approximate height of a human being. It can be seen from the results in Table 4 that the nights are riskier than the days in all seasons (54% in spring and winter, 72% in summer) but in autumn this trend is reversed, with 99% of the hours of risk occurring during the day. The main factors are those related to dilution (or its absence): wind speed and solar radiation at null levels. It should be noted that the temperature in the hours at risk of inhalation of gaseous Hg can be as low as –4 °C in winter, which is consistent with the idea expressed above that it is the dilution processes (or their absence) that most decisively influence

in the creation of periods of risk for the inhalation of gaseous Hg.

Once we had identified the micrometeorological conditions in which there was a risk, we proceeded to identify the extent of this risk in space. Profile 2 shows that the extent of the area affected by an emission source is independent of its importance in terms of absolute emissions, with the area not extending during the daytime period beyond 100 metres from the location of the source (Fig. 5). In the night, however, the extent of the affected area can reach more than 200 metres around the emission source (Fig. 8).

Likewise, it can be seen from profile 3 (Fig. 8) that the risk associated with the increase of TGM values as a result of the emissions of contaminated sediments from the Valdeazogues River is null, with values of 100 ng m$^{-3}$ not exceeded even under the worst micrometeorological conditions: i.e., summer nights without wind. The displacement of the Lepeltier curves in Fig. 8 shows that the cessation of wind during the night produces an increase of up to 33 ng m$^{-3}$ in the population of anomalous values, with only three values exceeding the WHO limit of 200 ng m$^{-2}$, while in profile 2 this increase reaches 178 ng m$^{-3}$ and, in this case, the

micrometeorological conditions prevalent at night mean that almost all of the anomalous population exceeds the WHO limit of 200 ng m$^{-3}$.





From the data discussed above one can estimate the extent of the risk areas in a basic way from the micrometeorological conditions. Two extreme cases for day and night, in summer and in the absence of wind, are represented in Fig. 9. It can be observed that during the night the affected area can reach almost 100% of the homes in Almadenejos, although it is necessary to remember that the results

of the MLRA indicated that only 11.34% of the hours of the studied year presented a risk of exceeding the WHO limit for chronic exposure (200 ng m$^{-3}$).

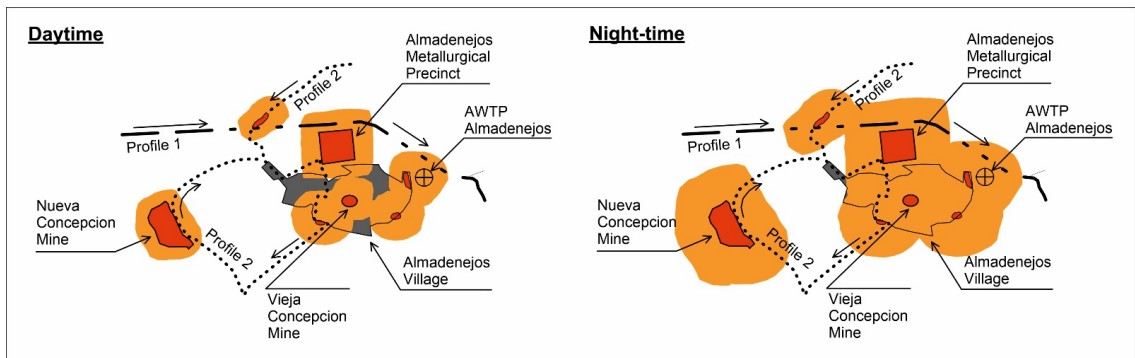

**Figure 9: Extent of areas with Hg inhalation risk in the worst scenarios (orange) in relation to the different source areas (red), during**
**daytime (left) and night-time (right).**

## 4 Conclusions

The study of transfer pathways of gaseous mercury in a mining-related environment has shown higher TGM levels at lower heights during nocturnal hours, relatively homogeneous and low levels during diurnal hours, and brief inversion periods during dawn and dusk.

Major sources act as constant suppliers of gaseous mercury to the diurnal mixing cell, while minor sources play an important role in mercury dispersion during nocturnal hours.

Vertical transferences occur preferentially during diurnal hours, while horizontal transferences predominate during nocturnal hours. The monitoring strategy provided sufficient data to delineate spatial and temporal risk areas. This monitoring work established the following as minimum data to be obtained in any given area affected by gaseous Hg emissions:

- identification of emission sources, with special emphasis on location and importance

- collection of data in a vertical transect at a fixed point during all seasons of the year

- collection of GEM data in horizontal transects that may include a combination of background and anomalous GEM values in its path, as well as day and night values.

A significant statistical treatment must be added to this TGM data acquisition strategy. It is proposed here that models should be

established using MLRA in order to allow the estimation of the times of risk based on past or expected micrometeorological data, without the need to re-measure after performing the risk assessment.

The results of this risk assessment show that nights are riskier than days in all seasons (54–72% in winter, spring and summer) but in autumn 99% of the higher-risk hours are diurnal. The main factors involved in the creation of periods of risk are those related to dilution (or its absence), e.g., wind speed and solar radiation at null levels. The extent of the affected area is independent of the

importance of the source in terms of absolute emissions, with the affected area not extending more than 100 metres from the location of the source during the daytime period and 200 metres in the night-time. The worst scenario produced an affected area that covered almost the entire town of Almadenejos, although these risk conditions only represent 11.34% of the hours in an annual period.



**Authors contribution**

The manuscript was written through contributions of all authors. JME and RN acquired all experimental data. JME and AMC applied
MLRA analysis to the dataset. JME, PHH and AMC produced the probabilistic Lepeltier graphs. All co-authors contributed to the
writing of the manuscript.

**Competing interests**

The authors declare that they have no conflict of interest.

**Acknowledgements**

This work was partly supported by projects CTM2012-33918 and CGL2015-67644-R, funded by the Spanish Ministry of Economy
and Competitiveness. We thank the town hall of Almadenejos (Ciudad Real) for allowing the use of their wastewater treatment
plant. Dr. Neil Thompson revised the English style.

**Authors contribution**

Financial support. This study was conducted under the CGL2015-67644-R project, funded by the Spanish Ministry of Economy and
Competitiveness.

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





**Tables.**

Table 1. Statistical summary of TGM levels at different heights (3, 2 and 0.5 metres) and total gradient (3–0.5 m), upper gradient (3–2 m) and lower gradient (2–0.5 m). All TGM data are in ng m$^{-3}$.

|  | N | Maximum | Minimum | Average |
|---|---|---|---|---|
| **All data** | | | | |
| TGM (3 m) | 6518 | 3012 | 2 | 125 |
| TGM (2 m) | 6518 | 2807 | 1 | 111 |
| TGM (0.5 m) | 6510 | 1971 | 1 | 102 |
| **Autumn** | | | | |
| TGM (3 m) | 1570 | 2378 | 2 | 134 |
| TGM (2 m) | 1570 | 1544 | 1 | 116 |
| TGM (0.5 m) | 1570 | 1360 | 1 | 104 |
| **Winter** | | | | |
| TGM (3 m) | 1222 | 1056 | 5 | 93 |
| TGM (2 m) | 1222 | 813 | 5 | 79 |
| TGM (0.5 m) | 1221 | 525 | 4 | 70 |
| **Spring** | | | | |
| TGM (3 m) | 1589 | 1506 | 7 | 93 |
| TGM (2 m) | 1589 | 837 | 5 | 80 |
| TGM (0.5 m) | 1582 | 819 | 4 | 75 |
| **Summer** | | | | |
| TGM (3 m) | 2137 | 3012 | 7 | 160 |
| TGM (2 m) | 2137 | 2807 | 7 | 147 |
| TGM (0.5 m) | 2137 | 1971 | 7 | 137 |
| ***Almadén** (Esbrí et al., 2016)* | | | | |
| *TGM Autumn* | *2025* | *281* | *0.8* | *23* |
| *TGM Winter* | *1159* | *122* | *0.8* | *13* |
| *TGM Spring* | *1067* | *280* | *3.1* | *23* |
| *TGM Summer* | *2019* | *687* | *2.5* | *52* |





Table 2. Statistical summary of gradients: total gradient (3–0.5 m), upper gradient (3–2 m) and lower gradient (2–0.5 m). All TGM
data are in ng m$^{-3}$.

|  | N | Maximum | Minimum | Average |
|---|---|---|---|---|
| **All data** | | | | |
| Gradient (3–0.5 m) | 6510 | 398 | −1270 | −23 |
| Gradient (3–2 m) | 6510 | 578 | −1066 | −9 |
| Gradient (2–0.5 m) | 6518 | 566 | −1280 | −14 |
| **Autumn** | | | | |
| Gradient (3–0.5 m) | 1570 | 398 | −1166 | −30 |
| Gradient (3–2 m) | 1570 | 578 | −756 | −12 |
| Gradient (2–0.5m) | 1570 | 566 | −1280 | −18 |
| **Winter** | | | | |
| Gradient (3–0.5 m) | 1221 | 164 | −531 | −23 |
| Gradient (3–2 m) | 1221 | 160 | −484 | −9 |
| Gradient (2–0.5 m) | 1222 | 468 | −400 | −14 |
| **Spring** | | | | |
| Gradient (3–0.5 m) | 1582 | 171 | −691 | −18 |
| Gradient (3–2 m) | 1582 | 263 | −559 | −6 |
| Gradient (2–0.5 m) | 1589 | 149 | −789 | −12 |
| **Summer** | | | | |
| Gradient (3–0.5 m) | 2137 | 371 | −1270 | −22 |
| Gradient (3–2 m) | 2137 | 433 | −1066 | −10 |
| Gradient (2–0.5 m) | 2137 | 318 | −801 | −13 |





Table 3. Predictor coefficients resulting from a multiple linear regression analysis (MLRA). The main predictors by season are shown in bold type. Abbreviations: Temp: outside temperature; Hum: outside humidity; WindSp.: Wind speed; Bar.Pres.: Barometric pressure; SolarRad.: Solar radiation; n.c.: not considered in the MLRA.

| | Constant | Temp | Hum | WindSp | Bar.Pres | Rain | SolarRad | $r^2$ |
|---|---|---|---|---|---|---|---|---|
| **Grad tot** | | | | | | | | |
| SUMMER | 0.010 | 0.112 | n.c. | **0.143** | 0.034 | -0.022 | 0.079 | 57.4 |
| SPRING | 0.001 | n.c. | 0.087 | 0.161 | −0.047 | 0.015 | **0.215** | 77.6 |
| AUTUMN | 0.000 | −0.068 | 0.043 | 0.164 | −0.101 | n.c. | **0.175** | 83.9 |
| WINTER | 0.001 | 0.127 | 0.092 | 0.165 | n.c. | 0.036 | **0.172** | 74.8 |
| **Grad inf** | | | | | | | | |
| SUMMER | 0.008 | **0.101** | −0.021 | **0.1** | −0.001 | n.c. | 0.056 | 85.8 |
| SPRING | 0.002 | n.c. | 0.063 | 0.133 | −0.037 | 0.013 | **0.156** | 85.0 |
| AUTUMN | 0.000 | −0.072 | n.c. | **0.114** | −0.054 | n.c. | 0.112 | 91.5 |
| WINTER | 0.000 | 0.083 | 0.091 | 0.117 | −0.045 | n.c. | **0.144** | 84.3 |
| **Grad sup** | | | | | | | | |
| SUMMER | 0.014 | 0.001 | n.c. | **0.07** | 0.015 | 0.002 | 0.052 | 84.1 |
| SPRING | 0.000 | n.c. | 0.044 | 0.067 | −0.025 | 0.032 | **0.121** | 78.6 |
| AUTUMN | 0.000 | n.c. | 0.059 | 0.094 | −0.084 | 0.038 | **0.117** | 82.9 |
| WINTER | 0.000 | 0.068 | n.c. | **0.097** | 0.034 | 0.03 | 0.073 | 78.3 |
| **TGM at 2 m. height** | | | | | | | | |
| SUMMER | −0.035 | −0.101 | 0.144 | **−0.236** | −0.139 | −0.052 | −0.197 | 82.4 |
| SPRING | −0.002 | −0.114 | −0.108 | **−0.345** | −0.065 | −0.085 | −0.282 | 81.6 |
| AUTUMN | −0.008 | 0.242 | 0.038 | −0.269 | −0.079 | −0.125 | **−0.390** | 84.5 |
| WINTER | 0.000 | −0.120 | −0.088 | −0.287 | 0.158 | −0.027 | **0.307** | 79.3 |





Table 4. Micrometeorological conditions for TGM outdoor values to exceed the WHO limit for chronic exposure of 200 ng m$^{-3}$.

Abbreviations: T: temperature; RH: relative humidity; WindSp: wind speed; and SR: solar radiation.

| | | T (ºC) min | RH (%) min | Windsp (m s$^{-2}$) max | SR (W m$^{-2}$) max | Time (h) N |
|---|---|---|---|---|---|---|
| Summer | Night | 11.5 | 29.2 | 1.9 | 0 | 204 |
| | Day | 10.5 | 33.5 | 1.7 | 0.2 | 93 |
| Autumn | Night | 22.8 | | 0 | 0 | 3 |
| | Day | 7.8 | | 3.4 | 0.5 | 280 |
| Winter | Night | −4.1 | | 0.3 | 0 | 155 |
| | Day | −4.2 | | 0.1 | 0.2 | 19 |
| Spring | Night | 7.7 | | 1.3 | 0 | 155 |
| | Day | 9.5 | | 2.3 | 0.2 | 85 |