# Peer review of "4D dispersion of total gaseous mercury derived from a mining source: identification of criteria to assess risks related with high concentrations of atmospheric mercury."

_Atmospheric Chemistry and Physics, 2019_

## Referee Comment (RC1) · Anonymous Referee #1 · 6 Apr 2020

General comments: The manuscript "4D dispersion of total gaseous mercury derived from a mining source: identification of criteria to assess risks related with high concentrations of atmospheric mercury" by Esbri et al. discusses criteria and a minimum amount of information needed to efficiently characterize Hg contaminated site as a result of past mining activities. The authors suggest a novel monitoring design and evaluate it based on results obtained during measurement campaigns in the Almaden mercury mining districts. Overall, the manuscript brings new insights into specific pathways of Hg at contaminated sites, as well as the methodology to determine risks associated
with it. The paper is well written and structured, including visualizations, statistical treatment and the interpretation of the results. However, there are some parts of the manuscript that are a bit unclear in its present form and need to be revised and simplified, respectively. To this end, in the following they are some specific comments and suggestions to improve the quality of this work. Specific comments: - Abstract: In its present form the abstract contains too many details, the second and third paragraphs in particular. It is suggested to rewrite it, focusing on the main outcomes of this work, e.g. relevant criteria and data needs for characterization of contamination and associated risks in the spatio-temporal context. - Line 46: Revise the definition of TGM - Line 64: "altitudes in the range 500-11,000 metres from background and contaminated locations;..." . Not clear. Revise and support with some references. - Lines 73-75: It is not clear what is meant by "Risk assessment", "...worst theoretical conditions..." and "...the worst-case scenario...". Revise and provide relevant details. - Lines 80-88: Not clear how the mentioned reference (Deng et al., 2016) is linked with the rest of the paragraph. It is also suggested to shorten and simplify this whole paragraph. - Line 97: Provide more details on the "exhaustive identification" of sources in the study area. By what means these sources were identified? - In Lines 100-103 emission sources in the study area are ranked according to their importance. Based on what criteria? - Line 109: Check if coordinates of AWTP are written in a correct format - Line 115: I suggest leaving out the sentence starting with "This situation gives..." - Lines 242-243: How were the background locations defined and separated from the rest? Technical corrections: - Line 120: Check values indicated in brackets for Lower and Upper Gradient - Lines 174-177: In Figure 2 there are no A, B and C mentioned in the text - Page 19: Location should be mentioned in Table 2 caption - Figure 5: units are not shown for scale bar in Profile 1 and Profile 3, respectively - Figure 9: scale bar is missing

---

## Author Comment (AC1) · 17 Apr 2020

General comments: The manuscript "4D dispersion of total gaseous mercury derived from a mining source: identification of criteria to assess risks related with high concentrations of atmospheric mercury" by Esbri et al. discusses criteria and a minimum amount of information needed to efficiently characterize Hg contaminated site as a result of past mining activities. The authors suggest a novel monitoring design and evaluate it based on results obtained during measurement campaigns in the Almaden mercury mining districts. Overall, the manuscript brings new insights into specific path-

ways of Hg at contaminated sites, as well as the methodology to determine risks associated The paper is well written and structured, including visualizations, statistical treatment and the interpretation of the results. However, there are some parts of the manuscript that are a bit unclear in its present form and need to be revised and simplified, respectively. To this end, in the following they are some specific comments and suggestions to improve the quality of this work. Specific comments: - Abstract: In its present form the abstract contains too many details, the second and third paragraphs in particular. It is suggested to rewrite it, focusing on the main outcomes of this work, e.g. relevant criteria and data needs for characterization of contamination and associated risks in the spatio-temporal context. Done. We have rewrite second and third paragraph to accomplish reviewer suggestion: The vertical profiles revealed that higher Total Gaseous Mercury concentrations are present at lower altitude during nocturnal hours and at higher altitude at dawn and dusk. Horizontal profiles showed that the background values were close to 6 ng m–3 except in the spring months, when they rose to 13 ng m–3 and increased the area affected by mercury emissions to more than 4 km around the mining and metallurgical sites. On a daily basis the most important process involved in gaseous mercury movements is the mixing layer, which begins in the early morning and finishes at nightfall. Vertical transferences are predominant when this process is active, i.e., in all seasons except winter, while major sources act as constant suppliers of gaseous Hg to the mixing cell, thus producing Hg deposition at dusk. Conversely, horizontal transferences prevail during the hours of darkness and the main factors are major and minor sources, solar radiation, wind speed and topography. The study has shown that it is important: i) to identify the sources; ii) to get data about Hg movements in vertical and horizontal directions; iii) to extend the measurements over time in a sufficiently representative way, both daily and seasonally; iv) to determine the different populations of data to establish the background levels, this work proposes the use of Lepeltier graphs to do it. In terms of risk assessment, and based on the model constructed to infer atmospheric Hg concentrations based on micrometeorological parameters, the nights carry greater risk than the days in all seasons (54%

in spring and winter, 72% in summer) except in autumn, when 99% of the hours of risk occurred during the day. The main factors involved in the creation of high-risk periods are those related to dilution (or its absence): namely wind speed and solar radiation at null levels. The extent of the area affected by an emission source is independent of its importance in terms of absolute emissions. The affected zone did not extend beyond 100 metres from the location of the source during the daytime period and 200 metres in the night-time. Under the worst micrometeorological conditions, it was predicted that the affected area would cover almost the entire town of Almadenejos, although these risk conditions only represent 11.34% of the hours in an annual period. - Line 46: Revise the definition of TGM Done. Now the sentence is "GEM and RGM species constitute 'total gaseous mercury' (TGM)". - Line 64: "altitudes in the range 500-11,000 metres from background and contaminated locations;. . ." . Not clear. Revise and support with some references. Done. The sentence is now as follows: Most of the available information on this topic is on a kilometric scale, at high altitudes in the range 500–11,000 metres from background and contaminated locations (Slemr et al., 2018; Weigelt et al., 2016); - Lines 73-75: It is not clear what is meant by "Risk assessment", ". . .worst theoretical conditions. . ." and ". . .the worst-case scenario. . .". Revise and provide relevant details. Done. The sentence is now as follows: Risk assessments of areas with anthropic contamination of gaseous Hg are often carried out with scarce data, often corresponding to short periods of time, and these do not provide a representative view of the day-night contrast or the seasonality, not even at the level of hot and cold or dry and wet seasons (depending on the location of the case study). We have conducted studies based on sampling times selected in the worst theoretical conditions, with higher expected emission rates enhanced by temperature and solar radiation, with the aim of identifying the worst-case scenario in summer days without winds in a mining site in Almadén (Martinez-Coronado et al., 2011), in a mining complex in Mount Amiata-Italy (Vaselli et al., 2013) in a chloralkali plant in Tarragona (Esbrí et al., 2015), in a chloralkali plant in Romania (Esbri et al., ; 2018a) and in a period of time with higher Hg metallurgical works in Almadén (Tejero et al., 2015), the evaluation

of background conditions (Higueras et al., 2014), and comparison of the worst and best scenarios (Higueras et al., 2013). - Lines 80-88: Not clear how the mentioned reference (Deng et al., 2016) is linked with the rest of the paragraph. It is also suggested to shorten and simplify this whole paragraph. Thanks for the suggestion, but we think it is important to put into perspective the argument that it is necessary to rethink the representativeness of the data when doing a mercury-related risk analysis. In many cases sampling (o monitoring?) is not performed (o carried out) in the worst conditions due to ignorance, since the variations in environmental concentrations of mercury can be very large in space and time and may be due to local reasons, not predictable based on the scientist's previous experience. For these reasons, we think that cited a reference like Deng et al (among many others), is relevant to support the idea. – Line 97: Provide more details on the "exhaustive identification" of sources in the study area. By what means these sources were identified? Done. The sentence is now as follows: In this work we have tried to obtain the minimum information necessary about the emission, transport and deposition of atmospheric mercury to ensure the representativeness of such data with a minimum cost in terms of effort and money. Before designing the sampling locations, an exhaustive identification of the Almadenejos emission sources, represented in red in Fig. 1, was carried out with a Lumex RA-915M equipment in mobile monitoring mode using a car to cover the entire area. - In Lines 100-103 emission sources in the study area are ranked according to their importance. Based on what criteria? Done. A new sentence has been added: The importance of the sources has been stablished if the average concentrations are below 200 ng m-3 (low importance), in the range of 200-1000 ng m-3 (medium importance) or up to 1000 ng m-3 (high importance). - Line 109: Check if coordinates of AWTP are written in a correct format Checked. - Line 115: I suggest leaving out the sentence starting with "This situation gives. . ." Yes, we have deleted this sentence. - Lines 242-243: How were the background locations defined and separated from the rest? Background values were determined using Lepeltier graphs. Technical corrections: - Line 120: Check values indicated in brackets for Lower and Upper Gradient Yes, we have checked and they

are correct. - Lines 174-177: In Figure 2 there are no A, B and C mentioned in the text Yes, the reference to the figure were incorrect. We have deleted A, B and C in the text. – Page 19: Location should be mentioned in Table 2 caption Done. We have changed the sentence: Table 1. Statistical summary of TGM levels at different heights (3, 2 and 0.5 metres) and total gradient (3–0.5 m), upper gradient (3–2 m) and lower gradient (2–0.5 m) in Almadenejos WWTP. All TGM data are in ng m–3. - Figure 5: units are not shown for scale bar in Profile 1 and Profile 3, respectively Yes, the scale was missing. We have revised the figure, adding these scales. - Figure 9: scale bar is missing Done.

**PROFILE 1**

Height (m.a.s.l.)

Valdeazogues R.   AMMP   Valdeazogues R.

0  1  2 km

Spring          N=7

Summer          N=4

Autumm          N=4

Winter          N=9

GEM (ng m⁻³)

**PROFILE 2**

Height (m.a.s.l.)

CR   AMMP   CR   NCM

0          0.5          1 km

Spring          N=10

Summer          N=5

Autumn          N=4

Winter          N=10

GEM (ng m⁻³)

**PROFILE 3**

Height (m.a.s.l.)

Almadenejos          Gargantiel
         Valdeazogues R.   Gargantiel R.

0  1  2 km

Spring          N=10

Summer          N=4

Autumn          N=4

Winter          N=9

GEM (ng m⁻³)

**Fig. 1.** Figure 5 revised

[Figure]

**Fig. 2.** Figure 9 revised

---

## Short Comment (SC1) · 12 May 2020

The manuscript studies the alternatives that exist to make monitoring works in an area contaminated with anthropogenic gaseous mercury. It recommends measurements at different heights, over significant transects and the repetition of these measurements over time. Although the results seem to be appropriate for a complex area such as the one they have chosen as study area, the effort involved in obtaining this minimum number of data is great and perhaps could be simplified if a previous study were made of the most important factors involved in the local cycle of mercury, in a short period.

[Figure]

In any case, the manuscript presents a monitoring option that seems to offer very significant data and that could be applicable to any contaminated area. Lines 44-47. Definition of TGM include wrongly particle-bound mercury fraction. Revise it. Lines 51-52. If water is included in this transfer pathways, what about sediments? Line 61. Again sediments are missing... Line 100. Explain what are the sources of medium importance, polluted wastes? Ore outcrops? Lines 128-131. Then there is no soil data, why? Line 147. Add a comma in 3,650 and unifies the way the figures are represented throughout the manuscript Lines 181-183. Add a reference to support this sentence. Line 210. There is no reference in the methodology section to soil temperature measurements, explain this. Line 250. Something is missed in the top of the figure, in the scale bars Line 287. In Figure 7, is it possible to separate transitional populations in the spring charts of profiles 1 and 3? Lines 307-312. The topographic profile is not enough to understand this, how are the river valleys? open or narrow? what is the difference in heights from the nearby mountains? and the slopes? Line 375. Indicate in the figure the inhabited area where risk from chronic exposure may occur Line 515. Unify the decimals in the numbers

---

## Author Comment (AC2) · 19 May 2020

The manuscript studies the alternatives that exist to make monitoring works in an area contaminated with anthropogenic gaseous mercury. It recommends measurements at different heights, over significant transects and the repetition of these measurements over time. Although the results seem to be appropriate for a complex area such as the one they have chosen as study area, the effort involved in obtaining this minimum number of data is great and perhaps could be simplified if a previous study were made of the most important factors involved in the local cycle of mercury, in a short

period. In any case, the manuscript presents a monitoring option that seems to offer very significant data and that could be applicable to any contaminated area. Lines 44-47. Definition of TGM include wrongly particle-bound mercury fraction. Revise it. Done. The new version explains this with the following sentence: GEM and RGM together these species constitute 'total gaseous mercury' (TGM). Lines 51-52. If water is included in this transfer pathways, what about sediments? Thanks for the comment. In principle, we did not consider it important to include sediments in this introduction, since we were going to study mercury emissions in an urban environment, but it is true that the results have shown that sediments are an important source of emission to consider in the area of study. For these reasons, we have included it in the following sentence: Numerous Hg transfer pathways are involved in this cycle, and these include soil-atmosphere, soil-plant, plant-atmosphere, and water-atmosphere, and sediments-water, amongst others. Line 61. Again sediments are missing. . . It is difficult to find references including data about gaseous mercury emissions from sediments, so we have included sediments with a reference to emissions from salt marshes, which are not a perfect analogue, but which well expresses the idea of the timing of mercury emissions in a river context. The new sentence is as follows: A maximum emission during diurnal hours was described for soils (Zhu et al., 2015), mine materials (Eckley et al., 2011), waters (O'Driscoll et al., 2003), sediments (Sizmur et al., 2017) and snow (Maxwell et al., 2013), while forb leaf (Stamenkovic et al., 2008) and growing broad leaf (Fu et al., 2016) reach their minimum emission rates during diurnal hours. Line 100. Explain what are the sources of medium importance, polluted wastes? Ore outcrops? Done. The new sentence is as follows: In the town centre of Almadenejos there are four emission sources of medium importance (cinnabar wastes), while in the vicinity there is one of very high importance (MMP), one of high importance (Nueva Concepción mine), and two of low importance (a contaminated road running North of the town and the course of the Valdeazogues river, since it passes through the El Entredicho mine). Lines 128-131. Then there is no soil data, why? The monitoring strategy was designed to study the vertical flows of mercury from dispersed emission

sources in the urban area, not to study emissions from the soils of the Almadenejos treatment plant, which are not contaminated. For this reason, the possibility of monitoring the temperature and moisture of these soils was not considered necessary. Line 147. Add a comma in 3,650 and unifies the way the figures are represented throughout the manuscript Done. Lines 181-183. Add a reference to support this sentence. We have not fully understood this suggestion. The sentence has a first reference to the wastes that exist in the metallurgical complex, which could be referenced by the already mentioned Martinez-Coronado et al. (2011), and a second part that describes minor sources identified in this work, and that is why we do not refer to them. We have added the reference to the first part of the sentence. Line 210. There is no reference in the methodology section to soil temperature measurements, explain this. These soil temperature data are used as reference but do not belong to this work, but to a previous one in which a nearby soil was monitored to study its emissions. In the present work, these data are used as general trends that are expected to continue year after year without significant changes. Unfortunately, the work remains unpublished and we have not been able to reference it. Line 250. Something is missed in the top of the figure, in the scale bars Done. The figure has been edited by a previous suggestion of reviewer 1 to solve this problem. Line 287. In Figure 7, is it possible to separate transitional populations in the spring charts of profiles 1 and 3? Thanks for the suggestion. Although the trend seems to change, there is not a sufficient number of data to establish an anomaly threshold. Lines 307-312. The topographic profile is not enough to understand this, how are the river valleys? open or narrow? what is the difference in heights from the nearby mountains? and the slopes? Done. A new sentence explaining this has been added: It is necessary to emphasize that the topography of the study area consists of mountainous alignments of smooth slopes, typical of the Appalachian relief, with maximum differences of heights of 220 meters. Line 375. Indicate in the figure the inhabited area where risk from chronic exposure may occur Done. The manuscript has a revised version of the figure highlighting the inhabited area Line 515. Unify the decimals in the numbers Done.

[Figure]

**Fig. 1.**

[Figure]

**Fig. 2.**

---

## Short Comment (SC2) · 1 Jun 2020

Lorenzo Reyes-Bozo

lorenzo.reyes@uautonoma.cl

The ACP-2019-1107 manuscript entitled "4D dispersion of total gaseous mercury derived from a mining source: identification of criteria to assess risks related with high concentrations of atmospheric mercury", offers an alternative for the characterization of environments contaminated by anthropogenic mercury gas. The manuscript contains original work and will be a valuable addition to the literature since report data of mercury obtained in different spatial region and temporal time (daily and different seasonal period).

The authors have studied the extent to which monitoring work must be extended to obtain sufficiently representative data. Ensuring the data representativeness in geochemical work has always been a major challenge. Working on soil geochemistry, this representativeness is highly dependent on heterogeneity for the elements studied, spatial distribution patterns, and aspects related to sample preparation and analysis. The gaseous character of mercury and atmospheric dynamics complicate the achievement of this purpose, and for this reason the manuscript proposes as necessary the extension in time and space of the monitoring works to ensure the representativeness of the data and thus be able to build a dispersion model of gaseous mercury in the study area. This approach of minimal monitoring work to do represents the main novelty of the manuscript and is adequately presented by the authors. Instead, there are limitations to this approach. The authors have selected a study area with passive mercury emission sources that are almost exclusively dependent on meteorology. It may be one of the simplest cases to monitor, but if the sources are active (for example, a chlor-alkali industry) or the emission sources are modified (for example, by remediation work on contaminated soils or mining environments), the constructed model shows weaknesses to offer useful data in a risk analysis context. The authors must explain these weaknesses of the model built in the discussion section or/and in the conclusions section. This explanation may be accompanied by a list of adaptation needs or its possible immediate application to different scenarios of interest: mercury contamination by artisanal gold mining, active industrial emissions (chlor-alkali industry, zinc ore smelters, etc) or including natural emissions of volcanism-related origin. Another important aspect to consider by the authors is the possibility of adapting this monitoring strategy to feed sufficiently representative data to models of dispersion of gaseous pollutants (Calpuff, ISC-Aermod, others). The role of wet and dry deposition and particulate mercury in the local mercury cycle must also be better explained. There are some details in the introduction and a reference by the same authors studying the topic is cited, but there are no references in the text to this topic.

The manuscript deserves to be published after this minor revision based on its novelty,

presentation and quality of the data provided.

---

## Author Comment (AC3) · 1 Jun 2020

The ACP-2019-1107 manuscript entitled "4D dispersion of total gaseous mercury derived from a mining source: identification of criteria to assess risks related with high concentrations of atmospheric mercury", offers an alternative for the characterization of environments contaminated by anthropogenic mercury gas. The manuscript contains original work and will be a valuable addition to the literature since report data of mercury obtained in different spatial region and temporal time (daily and different seasonal period). The authors have studied the extent to which monitoring work must

be extended to obtain sufficiently representative data. Ensuring the data representativeness in geochemical work has always been a major challenge. Working on soil geochemistry, this representativeness is highly dependent on heterogeneity for the elements studied, spatial distribution patterns, and aspects related to sample preparation and analysis. The gaseous character of mercury and atmospheric dynamics complicate the achievement of this purpose, and for this reason the manuscript proposes as necessary the extension in time and space of the monitoring works to ensure the representativeness of the data and thus be able to build a dispersion model of gaseous mercury in the study area. This approach of minimal monitoring work to do represents the main novelty of the manuscript and is adequately presented by the authors. Instead, there are limitations to this approach. The authors have selected a study area with passive mercury emission sources that are almost exclusively dependent on meteorology. It may be one of the simplest cases to monitor, but if the sources are active (for example, a chloralkali industry) or the emission sources are modified (for example, by remediation work on contaminated soils or mining environments), the constructed model shows weaknesses to offer useful data in a risk analysis context. The authors must explain these weaknesses of the model built in the discussion section or/and in the conclusions section. This explanation may be accompanied by a list of adaptation needs or its possible immediate application to different scenarios of interest: mercury contamination by artisanal gold mining, active industrial emissions (chlor-alkali industry, zinc ore smelters, etc) or including natural emissions of volcanism-related origin.

Thank you for the comments concerning our manuscript. As explained in the previous paragraph, the problem that this work sets out to solve was ensuring the data representativeness in the monitoring of areas contaminated with gaseous mercury. As stated in the text, our research group has worked extensively on these characterization procedures, on many occasions detecting data gaps that left part of the local cycle of mercury unexplained or characterized. We often tried to perform the characterization in the worst possible theoretical conditions, but later we found that it is not possible to know the worst possible theoretical conditions in all scenarios. The manuscript proposes a method with minimum work to do to ensure this representativeness, but it is true that the constructed model is adapted to the case study, and in this area, meteorological data can be used to model gaseous mercury concentrations since in the area emissions are passive and dependent on variations in temperature, wind and solar radiation. A paragraph at the end of the discussion section has been added to explain this weakness: "This approach is applicable with little variation to any area affected by diffuse Hg emissions, but will require adaptations if Hg emissions are active, whether it is anthropogenic (mostly industrial) or natural (volcanic related). In these cases, the monitoring procedures must be extended to the emission processes, with the aim of incorporating these data into the built model. In this way, the model will also serve to foresee changes in emission rates, either due to changes in technology in industrial activity, or due to changes in emission patterns in natural processes."

Another important aspect to consider by the authors is the possibility of adapting this monitoring strategy to feed sufficiently representative data to models of dispersion of gaseous pollutants (Calpuff, ISC-Aermod, others).

This suggestion is very interesting for the future works. We believe that it goes beyond the main objective of the present work, the construction of a simple model based on correlations between parameters that allow the application of this methodology without much economic cost or learning time of the mentioned models. That is why we have not considered them in this work, although it could be an interesting future line of research.

The role of wet and dry deposition and particulate mercury in the local mercury cycle must also be better explained. There are some details in the introduction and a reference by the same authors studying the topic is cited, but there are no references in the text to this topic.

Dry deposition rates were published in a previous manuscript and seems not to be involved in the cycle of TGM in the area. Risk related with this solid material are more related with the incorporation to human trophic chain. We must take in consideration

that a large proportion of Hg appears as bound to humic acids, a Hg compound more available for crops and vegetables. We have added some details about PBM in the methodology section: "Previous data of PBM of the area has shown that emissions are related with creation of diurnal mixing layer while dry deposition rates (317 $\mu$g m-2 year-1) were in the order of other rural areas, and lower than urban areas (Esbri et al., 2018b)"

The manuscript deserves to be published after this minor revision based on its novelty, presentation and quality of the data provided.

---

## Short Comment (SC3) · 3 Jun 2020

This manuscript offers monitoring alternatives for contaminated areas that seem to offer very significant results in mining areas such as the chosen one. In the context of emission reductions required by the Minamata Convention, these procedures should offer valuable information about the evolution of the gaseous Hg concentration values in areas with real problems of risk for people. Among all the work presented in the manuscript, I am very interested in making transects that can be compared over time, both in daily cycles and at different seasons. The method seems to work well in the

chosen mining environment, but I wonder if it would offer meaningful information in an environment with less spectacular emissions, for example, in a bay entering sediments contaminated with cinnabar and native mercury droplets. For the application of this transect monitoring method, is prior identification of the emission sources essential? What phenomena could I register in this case? Otherwise, the manuscript is very well written, and there are only a few minor errors that may have already spotted in the comments above. To name the ones that seemed most striking to me, the term TGM is not well defined on line 47, on line 60 I don't understand the term "forb", and the weather station is unclear where it is in Figure 1.

---

## Author Comment (AC4) · 3 Jun 2020

This manuscript offers monitoring alternatives for contaminated areas that seem to offer very significant results in mining areas such as the chosen one. In the context of emission reductions required by the Minamata Convention, these procedures should offer valuable information about the evolution of the gaseous Hg concentration values in areas with real problems of risk for people.

Thanks for this suggestion. The new scenario generated after the approval of the

[Figure]

Minamata Convention and its ratification by 120 countries will mean a major change in the levels of Hg available in the environmental compartments. This expected reduction should be monitored, to assess the evolution of the process and assess the adoption of more restrictions if the desired objectives are not achieved. In this sense, our systematic monitoring approach should offer comparable results over time and significant conclusions. Considering the importance of this suggestion, we have decided to include in the abstract a short sentence that indicates this aspect: "Furthermore, these systematic monitoring strategies can offer significant information in the context of the Minamata Convention emission reduction scenario." In addition, this is also commented in the last paragraph of the discussion section, which is now: "This approach is applicable with little variations to any area affected by diffuse Hg emissions, but will require adaptations if Hg emissions are active, whether it is anthropogenic (mostly industrial) or natural (volcanic related). In these cases, the monitoring procedures must be extended to the emission processes, with the aim of incorporating these data into the built model. In this way, the model will also serve to foresee changes in emission rates, either due to changes in technology in industrial activity, due to changes in emission patterns in natural processes or changes in emissions rates derived of restrictions of Minamata Convention (UN, 2019)." And we have added a reference: United Nation (2019). Minamata Convention on Mercury. Available at http://www.mercuryconvention.org/Convention/Text/tabid/3426/language/en-US/Default.aspx (Last access, 03/06/2020)

Among all the work presented in the manuscript, I am very interested in making transects that can be compared over time, both in daily cycles and at different seasons. The method seems to work well in the chosen mining environment, but I wonder if it would offer meaningful information in an environment with less spectacular emissions, for example, in a bay entering sediments contaminated with cinnabar and native mercury droplets. For the application of this transect monitoring method, is prior identification of the emission sources essential? What phenomena could I register in this case?

The better situation is to know the location of the most important emission sources prior to design the route of the transects, although locations of temporary sources (still unknown) can be incorporated into them, such as flood events that bring sediments rich in cinnabar and mercury droplets. The main advantage of this method of transects in different periods of time is the rapid and low-cost obtaining of comparable information that serves to establish background levels and anomalous levels and their evolution in the different meteorological seasons, in the day/night or in occasional events such as flood events, tides or that considered to have an influence on the activation of mercury gas emissions. As in other situations, prior knowledge improves the effectiveness of the approach.

Otherwise, the manuscript is very well written, and there are only a few minor errors that may have already spotted in the comments above. To name the ones that seemed most striking to me, the term TGM is not well defined on line 47

Done. Now the term is defined as: GEM and RGM constitute 'total gaseous mercury' (TGM).

on line 60 I don't understand the term "forb"

It is not a common term, it does not correspond to a single plant, but to plants with herbaceous flowers.

, and the weather station is unclear where it is in Figure 1.

Done. We have added a detail in the sentence: "The location of this device (WGS84 30S 351714 E/4289255 N) is shown in Fig. 1, in the AWTP Almadenejos."

---

## Referee Comment (RC2) · Anonymous Referee #3 · 27 Aug 2020

This manuscript presents an experimental design using atmospheric Hg monitoring instruments to improve the characterization of a Hg point source over time and space. There is indeed some interesting data and discussion in the manuscript. Nonetheless, I do not recommend the manuscript for publication for several reasons:

1. I feel it is not ideally suited to ACP. It is written as a methods paper based on its experimental design to improve the source characterization across four dimensions. Thus, I would recommend it's submission for Atmospheric Monitoring Techniques, or another similar journal. I do not feel it has the necessary impact or scope for ACP.

[Figure]

2. Given the direction of the paper (such that it is delivered as a methods style paper). I also see some short-comings here. The authors are validly critiquing the need for more time representative studies rather than short "snap-shots" in time that are typically made when taking mercury measurements (especially mobile ones) at source sites using active monitoring instruments. Yet their own work does exactly this. 4 snap-shots spread across the 4 seasons (I assume there is only one profile in each season). Are the days they did their horizontal transects truly representative of the whole season? Why is this approach any better than taking a single snap shot and describing the meteorological conditions present during said snap shot? This is particularly so because the sampling along the profiles was by changing location for each new sample, thus time can play a role in the observed concentration differences and not only spatial variation. Indeed, the authors even mention and discuss this, but it means changes in the measured concentrations can be related to both space and time. This exact point was raised in a study by McLagan et al., (2018). This study used passive samplers concurrently deployed in high numbers across the source area and the time integrated samples (over week long or seasonal deployments) give much more relevant data to assess chronic exposure risk and longer-term trends. The concurrent deployments mean concentration variability is limited to spatial differences. This study is highly relevant to this manuscript and should be discussed in detail (not referenced at all).

3. There is a lot of discussion of mixing layer or boundary layer characteristics based on only the TGM data measured at 3 different heights in the vertical profiles to a maximum of 3 m. Can these large scale phenomena (generally hundreds of metres be described with any certainty based of TGM measurements at three heights extending to only 3 m? I am highly skeptical of this. This applies to this whole section 3.1.

4. The methods section is lacking details. There is nothing describing when the horizontal profiles where made (time of day, date) and there is also nothing on the number of profiles made in each season. Thus, I have to assume each profile was only driven once per season? Thus, 4 "snap-shots-in-time". Details of the sampling instrumenta-

tion are also severely lacking. We need more details on the specific setup of the Tekran 2537B and the Lumex RA-915M to define the exact species being sampled. Heated lines, filters, sampling duration? At least reference another paper whose setup was followed. Were there any external injections to test the quality of the internal calibration source?

5. Some of the writing is also very heavy and needs to be made more concise. Whole paragraphs are used at times to make a point that could be summarised in a sentence and many sentences are very long and convoluted.

Specific comments: Abstract: Abstract is far too long. 680 words. It is heavy reading, where it should be a clear and consise summary.

Lines 47-49: This sentence really sums up one of the problems with this article. The writing is at times very convoluted and could be improved by making sentences more consise. Here stating "PBM & RGM are deposited on local or regional scales" or "PBM & RGM are deposited nearer to source" is enough.

Lines 49-50: "Once Hg is being deposited" should be "Once Hg HAS BEEN deposited"

Line 54-58: long, convoluted and repetitive sentence.

Line 64: "Metric scale" metric is the system, using this word to describe metre scales is very confusing. State "on the scale of metres"

Lines 72-74: Break this into two sentences.

Lines 79-80: There have been more recent studies on this very topic using passive samplers see McLagan et al. (2018). This study is highly relevant to this manuscript. And here seasonal differences are compared and the longer-term nature of the sampling method is ideal for chronic exposure assessment. Although it cannot make any diurnal assessment. This study should be discussed in detail in this manuscript.

Line 92: "Secular" wrong use of this word. It describes not being associated with

religion. i do not know of another definition such as that being the intention of the authors.

Lines 93-95: This is not ok. This does not need a Wiki quote. People know what the four dimensions are. Just like them without the Wiki reference.

Line 192: "...TGM concentrations close to zero..." Please change this to simply "lower". At no point do these concentrations get close to zero. especially considering typical background concentrations are less than 2ng/m3.

Lines 198-199: It seems difficult to state with much confidence that higher concentrations at ground level mean greater deposition. These are not flux measurements as there is a lot of influence of wind. It might be possible to also expect the higher elevation sample to be higher in mercury. Enrichment at the surface, especially in low wind conditions could suggests a source at the ground with decreasing concentration with elevation being caused by dilution with the less enriched air above. It makes sense there is little difference between the sampling heights in the day because the winds mix the system and little difference can be observed.

Figure 3: This is a poor figure. Simply categorizing the data as high medium or low removes any quantitative assessment of the data. This could be vastly improved by taking the mean of the three height measurements for teach hourly time period and then plotting the residuals of each sampling height against time. Thus describing the magnitude of differences.

Figure 4: Instead of presenting typical days with these weather patterns, why not present the mean data (and the number of days described by this weather) for each meteorological condition. The goes to the very heart of the purpose of the manuscript – to eliminate "snap-shots-in-time" and give better time integrated data.

Figure 5: why is the data so much more noisy in spring and autime than winter and summer in profile 3? This could be an analytical issue.

Lines 265-268: Couldn't this easily be confirmed with river water and sediment samples at each river crossing site?

Lines 279-281: Are they though? there looks to be little if any differences in overall concentrations of these profiles particularly for background concentrations based on Figure 5.

Lines 290-292: This may well be the case, but the sampling methods chosen do not relay any information as to whether this is a random and very short term spike in concentration or a longer-term trend. The measurement is merely a "snap-shot-in-time", making it exceedingly difficult to produce any assessment of chronic exposures.

Lines 298-299: again this is a short-coming of the method and an example of a time-related change in concentration rather than simply a spatial related change.

Lines 299-301: of course it is because wind increases dilution - it blows concentrations away and the mix with surrounding air depleted in TGM more rapidly.

Line 301: Why are we now talking about GEM and not TGM? This simply switched. Consistency of terminology please.

Line 307: But Profile 3 certainly does have emissions sources. You only have to look at the large spikes in TGM concentrations. The authors really needed to have a control profile, without any sources (rivers or mines) to make such a statement.

Conclusions: This point form conclusions is a little strange.

REFERENCES: McLagan, D. S., Monaci, F., Huang, H., Lei, Y. D., Mitchell, C. P., & Wania, F. (2019). Characterization and quantification of atmospheric mercury sources using passive air samplers. Journal of Geophysical Research: Atmospheres, 124(4), 2351-2362.

---

## Author Comment (AC5) · 11 Sep 2020

Dear Editor(s), thanks very much for driving this interesting (although somehow too long, true?) process of revision of our manuscript, which has received very interesting and important inputs aimed to improve its final quality. These are the responses to the last questions raised by reviewers. Some of them are comments, maybe not needed any response, but please tell us to provide responses to them if you consider it necessary. We think that we have properly answered the important concerns of this reviewer, which are also sincerely acknowledged by us; we particularly thanks his/hers

positive comments on the quality of our work, and we are sure that the proposal of deriving the manuscript to another Journal can be interesting, but please consider the question of this work being a contribution to the Special Issue of this Journal devoted to the 2019 International Conference of Mercury as a Global Pollutant. This was our initial aim, and we got the go ahead of Prof. Ashu Dastoor, invited editor of this SI, for this idea. We have written our responses in red bellow the reviewer's comments, and we have indicated in blue the new texts added to the manuscript in the context of this revision. We have upload a supplement file with these colours differentation. This manuscript presents an experimental design using atmospheric Hg monitoring instruments to improve the characterization of a Hg point source over time and space. There is indeed some interesting data and discussion in the manuscript. Nonetheless, I do not recommend the manuscript for publication for several reasons: 1. I feel it is not ideally suited to ACP. It is written as a methods paper based on its experimental design to improve the source characterization across four dimensions. Thus, I would recommend it's submission for Atmospheric Monitoring Techniques, or another similar journal. I do not feel it has the necessary impact or scope for ACP. 2. Given the direction of the paper (such that it is delivered as a methods style paper). I also see some short-comings here. The authors are validly critiquing the need for more time representative studies rather than short "snap-shots" in time that are typically made when taking mercury measurements (especially mobile ones) at source sites using active monitoring instruments. Yet their own work does exactly this. 4 snapshots spread across the 4 seasons (I assume there is only one profile in each season). Are the days they did their horizontal transects truly representative of the whole season? Why is this approach any better than taking a single snap-shot and describing the meteorological conditions present during said snap-shot? This is particularly so because the sampling along the profiles was by changing location for each new sample, thus time can play a role in the observed concentration differences and not only spatial variation. Indeed, the authors even mention and discuss this, but it means changes in the measured concentrations can be related to both space and

time. This exact point was raised in a study by McLagan et al., (2018). This study used passive samplers concurrently deployed in high numbers across the source area and the time integrated samples (over week long or seasonal deployments) give much more relevant data to assess chronic exposure risk and longer-term trends. The concurrent deployments mean concentration variability is limited to spatial differences. This study is highly relevant to this manuscript and should be discussed in detail (not referenced at all). 3. There is a lot of discussion of mixing layer or boundary layer characteristics based on only the TGM data measured at 3 different heights in the vertical profiles to a maximum of 3 m. Can these large scale phenomena (generally hundreds of metres be described with any certainty based of TGM measurements at three heights extending to only 3 m? I am highly skeptical of this. This applies to this whole section 3.1. 4. The methods section is lacking details. There is nothing describing when the horizontal profiles where made (time of day, date) and there is also nothing on the number of profiles made in each season. Thus, I have to assume each profile was only driven once per season? Thus, 4 "snap-shots-in-time". Details of the sampling instrumentation are also severely lacking. We need more details on the specific setup of the Tekran 2537B and the Lumex RA-915M to define the exact species being sampled. Heated lines, filters, sampling duration? At least reference another paper whose setup was followed. Were there any external injections to test the quality of the internal calibration source? 5. Some of the writing is also very heavy and needs to be made more concise. Whole paragraphs are used at times to make a point that could be summarised in a sentence and many sentences are very long and convoluted. The authors think that the manuscript cover the main aims and scopes of the journal. The subject of this work is centered in field measurements, as one of the main subjects stablish by the journal in the webpage (https://www.atmospheric-chemistry-and-physics.net/about/aims_and_scope.html ). Also, this work has general relevance and must be taken in account to further studies to avoid the obtaining of conclusions based on brief or not sufficiently representative data. For all these reasons and for the quality of the manuscript, as it can be deduced from the other

three complete reviews received, and other three short comments; on these bases, we think that the manuscript is suitable to be published on this journal. In fact, the authors must say that they have selected the special issue from the ICGMP 2019, not specially the journal, thinking that the manuscript has novelty enough to be selected by the guest editor, Prof. Ashu Dastoor, for this special issue. In the point 2, the reviewer suggest that our work criticize and make a short snap-shots, but we think that there are a misunderstood on this suggestion, because this point is not sufficiently explained in the materials and methods section. As it can be seen in figure 5, the number of monitoring of each season appears as N in each graph, corresponding to a range of 4 to 10 monitoring in different days. We have added a brief text in the method section to explain this point: "Data acquisition was carried out during 24 different days for profile 1, 29 for profile 2 and 27 for profile 3 during the period between May 2014 and June 2015. Monitoring days tried to include two meteorological conditions: days of wind calms and days with regional winds". But the reviewer suggestion that these short number of monitoring days remains to be snap-shots seems right to us. In fact, this is the main difficult of geochemistry works, for instance, to make a geochemical atlas of our region, we take 908 soil samples from an area of some 80,000 km2, and we make a geochemistry map of elements distribution. We must assume that there is a distance between our representation of reality and reality itself and the objective of this study is to try to make that distance the minimum acceptable, based on real data. To reduce this distance there are two main approaches: randomizing the sampling or doing a sampling that takes into account the main factors of heterogeneity. In the proposed example of the atlas, we have dealt with it by choosing the samples based on the lithologies since these are the main source of heterogeneity in the soils of the region. In the research work corresponding to this manuscript, we have identified wind as the main source of heterogeneity, and it has been this parameter that has been used to choose the different monitoring days. It is mandatory to carry out these monitoring that look for spatial patterns of dispersion in similar time periods between them so that they can be comparable. In this sense, it is essential to take snap-shots that correspond

to comparable periods of the day, but we must remember that with these transects the spatial patterns of dispersion of GEM are sought, and that the temporal pattern is studied through continuous measurements during a whole year with the Tekran device. Reviewer suggest the use of passive samplers as a good solution to solve this problem and this is a significative suggestion that we have not taken into account when drafting the manuscript. We have tried to solve this in the introduction section, adding the following paragraph: "These objectives can be accomplish sufficiently using passive samples (McLagan et al., 2018), with clear advantages in its low cost and the easier application, especially in areas with access difficulties. Some uncertainties remain in this approach, most important of them is the Hg compounds that these passive samplers' uptake. This uncertainty can be important in the vicinity of industrial sites (for instance, chloralkali plants), where RGM can be in higher proportions". And in the results and discussion section a new paragraph has been added: "All datasets measured for these three profiles correspond to a period (11:00–14:00) of stability in terms of micrometeorological parameters, i.e., in the middle of the day. This approach is essential to ensure the comparability of the different transects, but it is a limitation in the temporal evolution of GEM contents throughout the day. The present work complements these daytime measurements with night-time ones, based on the daily evolution described in the area (Esbrí et al., 2016, Tejero et al., 2015), but it should be mentioned that there is an alternative to carry out these monitoring tasks using passive samplers (McLagan et al., 2018), which offer a greater time range. Their use as a substitute for these direct measures or in combination with them will undoubtedly result in higher representativeness of the data obtained. These measures during summer nights reported higher GEM levels in the surroundings of mining-related GEM sources, with levels more than two times higher in Almadén, for instance." Another point in the review is the role of the mixing layer in the Hg distribution of the investigated area. May be it is not the creation of the mixing layer itself, but the consequence of this creation in terms of winds at the monitoring heights (not more than 3 meters). Other authors have described this process as a main process involved

in pollutants concentrations in the nearby town of Puertollano (Adame et al., 2012). We add a comment in this sense to clarify this point: "This phenomenon must be due to the confluence of three micrometeorological factors: high temperatures and solar radiation coincidental with low relative humidity values, which combine to increase the intensity of the formation of the mixing layer during the day, that has the consequence of an increment of wind speed in the investigated area." Adame, J. A., Notario, A., Villanueva, F., & Albaladejo, J. (2012). Application of cluster analysis to surface ozone, NO 2 and SO 2 daily patterns in an industrial area in central-southern Spain measured with a DOAS system. Science of the Total Environment, 429, 281-291. In the point 4, the suggestion to give more details about what gaseous Hg specie has been measured with Tekran and Lumex was the most common discussion point in the previous reviews and short comments. We think that in the revised version has been solved satisfactorily with the aid of this suggestions. Specific comments: Abstract: Abstract is far too long. 680 words. It is heavy reading, where it should be a clear and concise summary. We have tried to summarize the abstract, we have this version in 562 words with all suggestion attended, not only of this reviewer, but also of the previous review and the short comments. "Mercury is a global pollutant that can be transported long distances after its emission by primary sources. The most common problem of gaseous Hg in the vicinity of anthropogenic sources is it presence in inorganic forms and in the gaseous state in the atmosphere. Risk assessments related to the presence of gaseous Hg in the atmosphere at these contaminated sites are often based on episodic and incomplete data, which do not properly characterize the Hg cycle in the area of interest or consider spatial or temporal terms. The aim of the work described was to identify criteria to obtain the minimum amount of data with the maximum meaning and representativeness in order to delimitate risk areas, both in a spatial and temporal respect. Data were acquired from May 2014 to August 2015 and included vertical and horizontal Hg measurements. A statistical analysis was carried out and this included the construction of a model of vertical Hg movements that could be used to predict the location and timing of Hg inhalation risk. A monitoring

strategy was designed in order to identify the relevant criteria and this involved the measurement of gaseous Hg in a vertical section at low altitude (i.e., where humans are present) and in horizontal transects to characterize appropriately the transport cycle of gaseous Hg in the lower layers of the atmosphere. The measurements were carried out over time in order to obtain information on daily and seasonal variability. The study site selected was Almadenejos (Ciudad Real, Spain), a village polluted with mercury related to decommissioned mining and metallurgical facilities belonging to the Almadén mercury mining district. The vertical profiles revealed that higher Total Gaseous Mercury concentrations are present at lower altitude during nocturnal hours and at higher altitude at dawn and dusk. On a daily basis the most important process involved in gaseous mercury movements is the mixing layer. Vertical transferences are predominant when this process is active, i.e., in all seasons except winter, while major sources act as constant suppliers of gaseous Hg to the mixing cell, thus producing Hg deposition at dusk. Conversely, horizontal transferences prevail during the hours of darkness and the main factors are major and minor sources, solar radiation, wind speed and topography. The study has shown that it is important: i) to identify the sources; ii) to get data about Hg movements in vertical and horizontal directions; iii) to extend the measurements over time in a sufficiently representative way, both daily and seasonally; iv) to determine the different populations of data to establish the background levels, this work proposes the use of Lepeltier graphs to do it. In terms of risk assessment, the nights carry greater risk than the days in all seasons except in autumn. The main factors involved in the creation of high-risk periods are those related to dilution (or its absence): namely wind speed and solar radiation at null levels. The results of this study highlight the possible importance of the relief in the distribution of gaseous mercury in the proximity of discrete sources. Furthermore, these systematic monitoring strategies can offer significant information in the Minamata Convention emission reduction scenario. Further studies, including a detailed topographic model of the area, are required in order to make precise estimations of the influence of this parameter, which appears in this study to be less important than the other factors but

is still appreciable." Lines 47-49: This sentence really sums up one of the problems with this article. The writing is at times very convoluted and could be improved by making sentences more concise. Here stating "PBM & RGM are deposited on local or regional scales" or "PBM & RGM are deposited nearer to source" is enough. We agree with the reviewer, we have tried to be concise and to include only relevant information in the manuscript, but we cannot always achieve this purpose because English is not our mother language, and we must to use a scientific reviewer (Dr. Neil Thompson, mentioned in the acknowledgements) to make our "spanglish" readable. We think that the revisor makes an excellent job with our way of writing, but perhaps along the writing process, the objective of being concise can be lost. We have tried to simplify sentences through the manuscript, following this suggestion. Lines 49-50: "Once Hg is being deposited" should be "Once Hg HAS BEEN deposited" Done. Line 54-58: long, convoluted and repetitive sentence. We agree with the reviewer, the sentence was hard to understand. Now it is as follow: "Results show that processes of Hg deposition and emission are included in a complex cycle with a large number of factors involved, mainly seasonality, vegetation coverage, temperature, solar radiation, relative humidity, diurnal atmospheric turbulence and the presence of Hg oxidants (Zhu et al., 2016)." Line 64: "Metric scale" metric is the system, using this word to describe metre scales is very confusing. State "on the scale of metres" Done. We have changed this and other previous of "kilometric scale" Lines 72-74: Break this into two sentences. Done. Lines 79-80: There have been more recent studies on this very topic using passive samplers see McLagan et al. (2018). This study is highly relevant to this manuscript. And here seasonal differences are compared and the longer-term nature of the sampling method is ideal for chronic exposure assessment. Although it cannot make any diurnal assessment. This study should be discussed in detail in this manuscript. We agree with the suggestion, but we think that this reference in the manuscript remains to be valid, because its meaning in the line of arguments about the importance of using representative data to make these statements. We think that the reference to the work of McLagan et al. (2018) can be added after this paragraph, to include another

valid approach to solve this, as an alternative of the proposed in the manuscript. We have included this text: "These objectives can be accomplish sufficiently using passive samples (McLagan et al., 2018), with clear advantages in its low cost and the easier application, especially in areas with access difficulties. Some uncertainties remain in this approach, most important of them is the Hg compounds that these passive samplers' uptake. This uncertainty can be important in the vicinity of industrial sites (for instance, chloralkali plants), where RGM can be in higher proportions." Line 92: "Secular" wrong use of this word. It describes not being associated with religion. i do not know of another definition such as that being the intention of the authors. We tried to solve this, but we see in Wikipedia that this term can be adequate if we read this definition of secular variation: "The secular variation of a time series is its long-term non-periodic variation (see Decomposition of time series)." Lines 93-95: This is not ok. This does not need a Wiki quote. People know what the four dimensions are. Just like them without the Wiki reference. We agree with this suggestion, but another previous reviewer suggests this. We have deleted this addition. Line 192: ". . .TGM concentrations close to zero. . ." Please change this to simply "lower". At no point do these concentrations get close to zero. especially considering typical background concentrations are less than 2ng/m3. Done. Lines 198-199: It seems difficult to state with much confidence that higher concentrations at ground level mean greater deposition. These are not flux measurements as there is a lot of influence of wind. It might be possible to also expect the higher elevation sample to be higher in mercury. Enrichment at the surface, especially in low wind conditions could suggests a source at the ground with decreasing concentration with elevation being caused by dilution with the less enriched air above. It makes sense there is little difference between the sampling heights in the day because the winds mix the system and little difference can be observed. This is not a study of vertical mercury fluxes from a contaminated surface (e.g., a polluted soil), but rather the vertical fluxes of Hg that came from nearby sources that were being monitored. In this sense, perhaps we should include these findings as suggestions since the vertical fluxes of Hg are not being quantified. We

have made changes in this regard to the text: "These positive differences between heights in terms of TGM suggest that mercury can remain accumulated at lower heights during the night, rising while the mixing layer is being created, and falling when this mixing layer disappears. These data could indicate that a diurnal cycle of emission and deposition is active in the studied area, and that deposition could be intense – especially at dusk – in the transitional hours between higher and lower winds." Figure 3: This is a poor figure. Simply categorizing the data as high medium or low removes any quantitative assessment of the data. This could be vastly improved by taking the mean of the three height measurements for teach hourly time period and then plotting the residuals of each sampling height against time. Thus describing the magnitude of differences. Yes, the original design of the figure was as the reviewer suggest, but we think that the meaning of the data provided was hard to understand, and we tried to simplify the figure in the same spirit as heat maps, commonly used nowadays. Some meaning has been lost with this simplification, but we think that the essential meaning of data to be discuss in the text is in the figure. Figure 4: Instead of presenting typical days with these weather patterns, why not present the mean data (and the number of days described by this weather) for each meteorological condition. The goes to the very heart of the purpose of the manuscript – to eliminate "snap-shots-in-time" and give better time integrated data. We have assumed that there were exceptional micrometeorological conditions that must be explained and that they are not sufficiently represented in the general data since their influence is diluted in the prevailing conditions. This is the sense of this figure. Figure 5: why is the data so much more noisy in spring and autime than winter and summer in profile 3? This could be an analytical issue. We think that the noise that the reviewer has seen is related with the changes in micrometeorological conditions in these transitional seasons. We have explained this effect in previous works, such as Esbri et al. (2016). Esbrí, J. M., Martínez-Coronado, A., & Higueras, P. L. (2016). Temporal variations in gaseous elemental mercury concentrations at a contaminated site: Main factors affecting nocturnal maxima in daily cycles. Atmospheric Environment, 125, 8-14. Lines

265-268: Couldn't this easily be confirmed with river water and sediment samples at each river crossing site? Yes, we have added a new reference of García-Ordiales et al. (2018) in this sense. Garcia-Ordiales, E., Higueras, P., Esbrí, J. M., Roqueñí, N., & Loredo, J. (2018). Seasonal and spatial distribution of mercury in stream sediments from Almadén mining district. Geochemistry: Exploration, Environment, Analysis, 19(2), 121-128. Lines 279-281: Are they though? there looks to be little if any differences in overall concentrations of these profiles particularly for background concentrations based on Figure 5. Yes, we propose Lepeltier approach to avoid personal interpretation based on box and whistler graphs. In soil geochemistry, this approach provides more precise information about background values and anomalous populations. Also, differences between them appears as more significative. We have worked with this approach in Higueras et al. (2003) Higueras, P.; Oyarzun, R.; Biester, H.; Lillo, J.; Lorenzo, S. (2003) A first insight into mercury distribution and speciation in the Almadén mining district, Spain. Journal of Geochemical Exploration, 80: 95-104. Lines 290-292: This may well be the case, but the sampling methods chosen do not relay any information as to whether this is a random and very short term spike in concentration or a longer-term trend. The measurement is merely a "snap-shot-intime", making it exceedingly difficult to produce any assessment of chronic exposures. We think that we have answered this misunderstood previously, but we insist that these profiles are not merely snap-shot, we have made these monitoring in the middle of the day, with a more stable wind condition, because is the unique way to have comparable data for all points considered. And is important to remember that the objective was to search spatial variations, not temporal variations, and a try to identify patterns of Hg distribution and factors. Chronic exposure must be assessed with secular data of a whole year, as it has been our research plan. Lines 298-299: again this is a short-coming of the method and an example of a timerelated change in concentration rather than simply a spatial related change. Sorry, we do not understand this comment. Does the review consider this as reiterative? Or invalid? Lines 299-301: of course it is because wind increases dilution - it blows concentrations

away and the mix with surrounding air depleted in TGM more rapidly. We agree. Line 301: Why are we now talking about GEM and not TGM? This simply switched. Consistency of terminology please. Yes, we have solved this problem with the previous suggestions of other reviewers. Line 307: But Profile 3 certainly does have emissions sources. You only have to look at the large spikes in TGM concentrations. The authors really needed to have a control profile, without any sources (rivers or mines) to make such a statement. We added "significant" sources to indicate that in this profile the source has very low capacity to emit Hg. It will be preferably to have a blank profile in the area, but in the Almadén mining district is impossible to find a profile like this. Centuries of mining exploitation and the dissemination of artisanal furnace to recover Hg from cinnabar make impossible the search of such blank profile. Conclusions: This point form conclusions is a little strange. REFERENCES: McLagan, D. S., Monaci, F., Huang, H., Lei, Y. D., Mitchell, C. P., & Wania, F. (2019). Characterization and quantification of atmospheric mercury sources using passive air samplers. Journal of Geophysical Research: Atmospheres, 124(4), 2351-2362.

Please also note the supplement to this comment:
https://acp.copernicus.org/preprints/acp-2019-1107/acp-2019-1107-AC5-supplement.pdf